# Extended embryo retention and viviparity in the first amniotes

**Baoyu Jiang** [1] ✉, **Yiming He**[1], **Armin Elsler** [2], **Shengyu Wang**[1], **Joseph N. Keating** [1], **Junyi Song**[1], **Stuart L. Kearns**[2] **& Michael J. Benton** [2]

The amniotic egg with its complex fetal membranes was a key innovation in vertebrate evolution that enabled the great diversification of reptiles, birds and mammals. It is debated whether these fetal membranes evolved in eggs on land as an adaptation to the terrestrial environment or to control antagonistic fetal–maternal interaction in association with extended embryo retention (EER). Here we report an oviparous choristodere from the Lower Cretaceous period of northeast China. The ossification sequence of the embryo confirms that choristoderes are basal archosauromorphs. The discovery of oviparity in this assumed viviparous extinct clade, together with existing evidence, suggests that EER was the primitive reproductive mode in basal archosauromorphs. Phylogenetic comparative analyses on extant and extinct amniotes suggest that the first amniote displayed EER (including viviparity).

The amniotic egg is very different from the anamniotic egg of extant amphibians, which lacks an eggshell and extraembryonic membranes. The amniotic egg consists of a suite of fetal membranes, including the amnion, chorion and allantois, as well as an external shell that can be either strongly mineralized (as in rigid-shelled eggs) or weakly mineralized (as in parchment-shelled eggs). The extraembryonic membranes enclose specific egg elements, regulate gas and fluid exchange between the egg and the external environment, store nutrients and collect waste[1–3].

Where and how the fetal membranes of the amniotic egg evolved has been debated, and two competing hypotheses have been proposed (Fig. 1). The conventional, 'terrestrial model'[2] is that the precursor to amniotes laid eggs on land, similar in many respects to the directly developing eggs of a variety of extant amphibians, and the fetal membranes were gradually acquired so that the egg could adapt to terrestrial environments by retaining water inside and allowing oxygen and carbon dioxide to pass through the eggshell. This widely accepted model has been challenged by the 'extended embryo retention model'[3–7], that the extraembryonic membranes appeared in the oviducts of the amniotic ancestor as specializations to control fetal–maternal interaction in association with extended embryo retention (EER). The EER model could occur with the embryo either in a post-neurula stage

(oviparity)[3] or with live bearing (viviparity)[6,7]. Among extant amniotes, turtles, crocodilians and birds generally lay eggs at an early developmental stage (non-EER oviparity), whereas most squamates (lizards and snakes) and mammals either display oviparity with EER or viviparity. Evolutionary studies based on extant amniotes give equivocal results about whether oviparity or viviparity arose first[8–13]. Circumstantial evidence for the EER model is the near absence of fossils of amniotic eggs before the Late Triassic period and the discovery of viviparity in many extinct amniotes as old as the Early Permian period[14–17]. Supporters of the terrestrial egg model note that EER is absent in archelosaurs, including chelonians, crocodiles and birds, as well as extinct dinosaurs, pterosaurs and their ancestors[14,18].

Whether the first amniote displayed EER or not is key to testing between the two models. As EER occurs widely among extant lizards and snakes (squamates) and mammals[3], exploring the occurrence of EER among oviparous primitive archosauromorphs is decisive to determine the developmental stage of the first amniotic egg (Fig. 1). In this study, we report an articulated embryo of the choristodere *Ikechosaurus* sp. inside a parchment-shelled egg. The ossification sequence of the embryo confirms that choristoderes are basal archosauromorphs. The shell structure reveals that the aquatic choristodere was oviparous and presumably came ashore to lay its eggs, like extant

[1]State Key Laboratory for Mineral Deposits Research, School of Earth Sciences and Engineering and Frontiers Science Center for Critical Earth Material Cycling, Nanjing University, Nanjing, China. [2]School of Earth Sciences, Life Sciences Building, Tyndall Avenue, University of Bristol, Bristol, UK. ✉e-mail: byjiang@nju.edu.cn

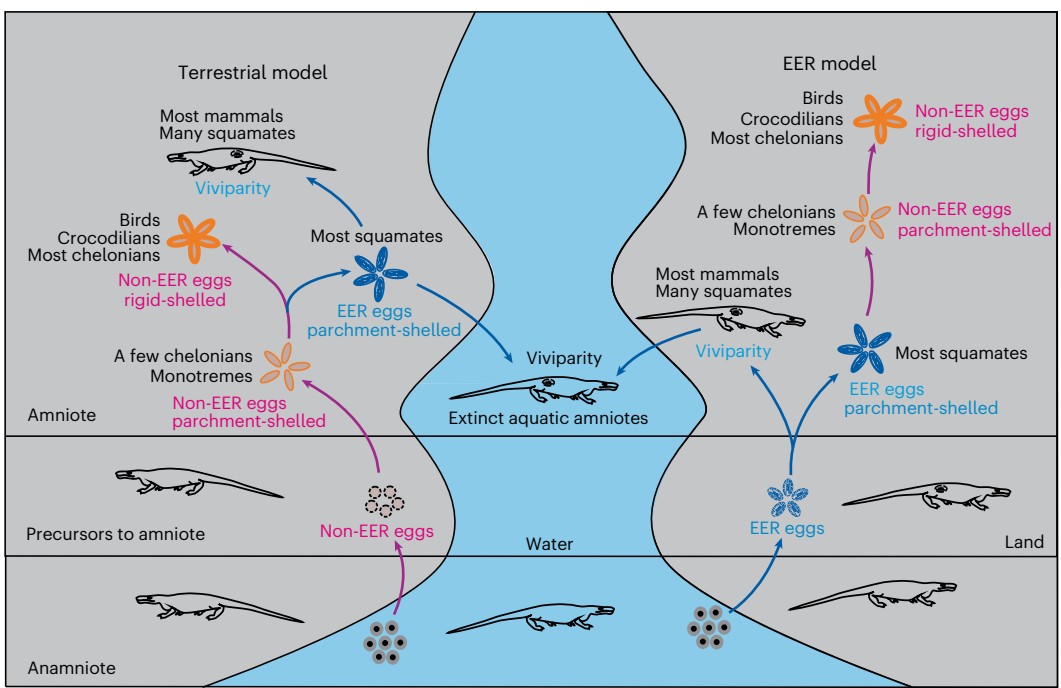

**Fig. 1 | The two competing theories for the evolution of the amniotic egg.** In the terrestrial model (left), non-EER oviparity (purple) was the primitive condition; oviparity with EER and viviparity (blue) evolved multiple times in amniotes. In the EER model (right), the evolutionarily labile reproductive mode of EER across oviparity to viviparity (blue) was primitive, while non-EER oviparity (purple) evolved multiple times in amniotes.

sea turtles and crocodilians. The specimen, together with previous evidence of viviparity in other taxa, demonstrates that an evolutionarily labile reproductive strategy across oviparity to viviparity existed in choristoderes, a basal clade of archosauromorphs, and potentially also in other various aquatic vertebrates of the past, such as mesosaurs, ichthyosaurs and sauropterygians. We run phylogenetic analyses on extant and extinct amniotes to test whether EER and viviparity are the ancestral conditions in Amniota.

## Results

### Viviparity and oviparity in choristoderes

Choristoderes are a clade of extinct diapsids that lived primarily in Laurasia from the Middle Jurassic period to the Early Miocene epoch (approximately 168–120 Myr). The gavial-like neochoristoderes were top predators in freshwater bodies, competing with contemporaneous crocodiles[19]. The new specimen (MES-NJU 57004) was collected from yellowish white, thinly laminated tuffaceous mudstone of the Lower Cretaceous Jiufotang Formation (Jehol Biota, approximately 125–120 Myr) in the Lamagou locality adjacent to Chaoyang City, western Liaoning, northeast China. Many choristoderes have been discovered in the Jiufotang Formation and the underlying Yixian Formation in western Liaoning, including the lizard-like *Monjurosuchus* and *Philydrosaurus*, the long-necked *Hyphalosaurus* and the neochoristoderan *Ikechosaurus*. Some specimens of these choristoderes are associated with eggs and embryos[19].

The new specimen is a small skeleton (approximately 102.73 mm long; Fig. 2a) that exhibits the typical pose of a vertebrate embryo: curving and with the head contacting the tail[17]. The skeleton is dorsoventrally flattened and exposed in ventral view covered by a thin layer of ferric oxide. Computed tomography (CT) scans reveal that the skeleton is nearly complete and all bony elements are articulated except for the distal end of the tail, which was slightly displaced (Fig. 3 and Extended Data Figs. 1 and 2). The embryo shows many diagnostic traits of *Ikechosaurus*: the snout is long, broad and flat in front of the

orbits and gradually tapers to about the midpoint along the snout; the interorbital bar is narrow; the jugal extends anteriorly to the midpoint of the lacrimal; the postorbital region is flared; the temporal openings lie largely above one another; and the parietal extends only about half way along the posterior edge of the upper temporal opening[20] (see the Supplementary Results for a detailed description of the skeleton). The fact that *Ikechosaurus* is the only neochoristoderan in the region supports this assignment[19].

The phylogenetic position of Choristodera in Amniota remains controversial, having been placed as a basal clade of archosauromorphs[21], a sister group of archosauromorphs, or basal to archosauromorphs and lepidosauromorphs[22]. The ossification sequence of the new embryo confirms that choristoderes are basal archosauromorphs (see Supplementary Results for evidence that the choristodere embryo is an archosauromorph). The high degree of ossification of the skeleton indicates that the unhatched embryo was in a late developmental stage. Ontogenetically, the animal was precocial or superprecocial[23]: its well-developed skull with sharp teeth suggests that it was ready to hunt and the relatively well-ossified pelvic girdle and hindlimbs that it could run and swim soon after hatching.

We identified traces of a parchment-shelled egg around the tiny skeleton, which is embedded in an incomplete, oval phosphate matrix, demarcated by a 0.43–0.50-mm-wide halo. The halo contains materials from both the phosphate matrix and the enclosing mudstone (Fig. 2d–g). The small size, embryonic pose and egg-shaped matrix prove that this is an embryo inside an egg. The outer edge of the halo is slightly meandering and locally folded. Loosely arranged, irregularly shaped, flake-like structures (around 50-μm thick by estimate) are locally present along the marginal area, surrounding pore-like structures (Fig. 2b,c). These features indicate that the halo is preserved eggshell, which is pliable and has a thin calcareous layer composed of flake-like shell units and many pores[24].

There are three types of amniotic eggs: membrane-shelled; parchment-shelled; and rigid-shelled[25,26]. A mineral layer is not

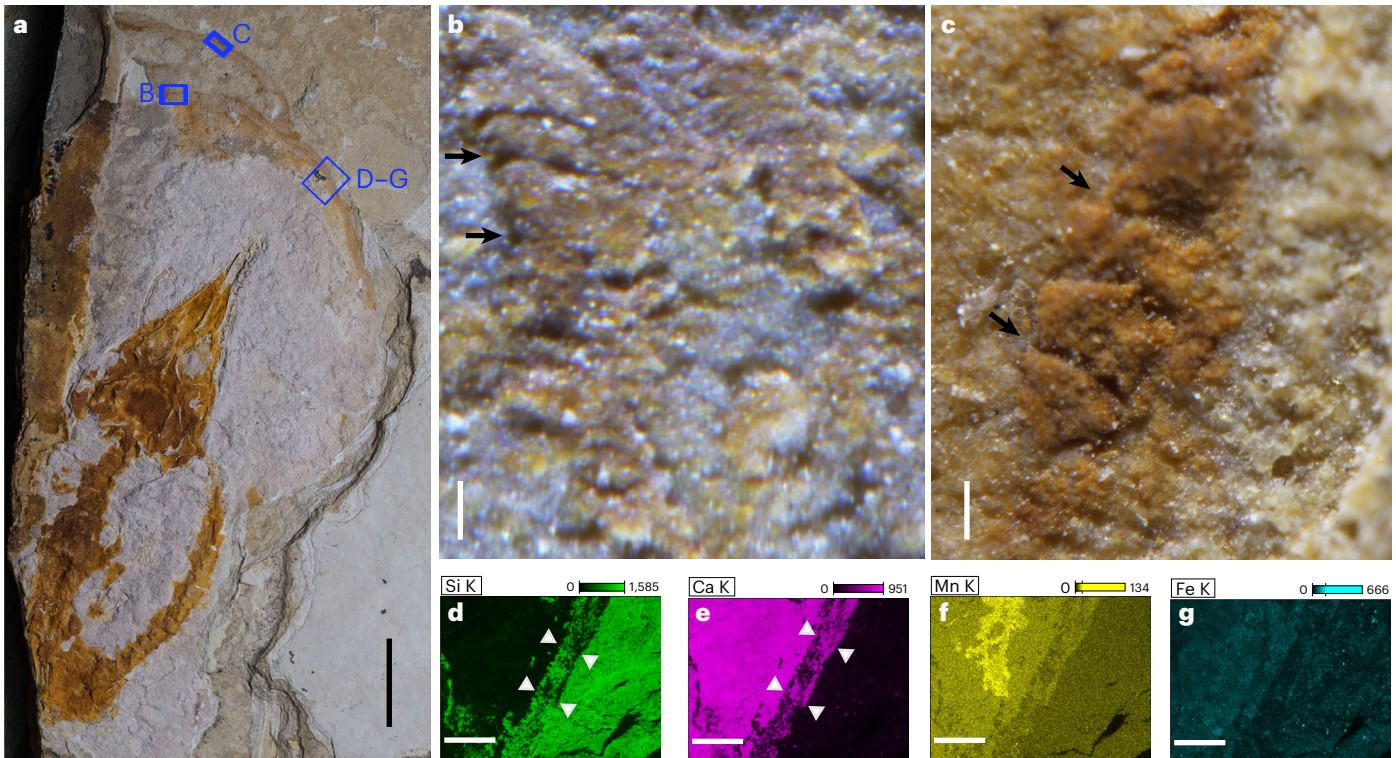

**Fig. 2 | Structure of the choristodere egg (MES-NJU 57004). a**, An overview of the embryo inside the egg. **b,c**, Photomicrographs show loosely arranged shell units and pores (black arrows) preserved along the marginal zone of the egg. **d–g**, Energy dispersive spectroscopy mapping of silica (**d**), calcium (**e**), manganese (**f**) and iron (**g**) showing the inferred eggshell (white triangles), which contains materials from both the phosphate matrix and the enclosing mudstone. Scale bars, 1 cm (**a**), 100 μm (**b,c**), 1 mm (**d–g**).

developed in membrane-shelled eggs (monotremes, a few squamates), is thinner than an organic layer in parchment-shelled eggs (a few chelonians and most squamates) and is well developed and thicker than the organic layer in rigid-shelled eggs (a few squamates, most chelonians, crocodilians and birds)[25]. It could be argued that the described choristodere specimen is an incomplete viviparous egg[27,28]. In extant squamates, the egg of viviparous species lacks the calcified layer but is enveloped with a very thin organic layer (commonly less than 10 μm), whereas oviparous squamates lay eggs with a calcified outer layer and a relatively thick organic layer (usually over 30 μm)[29,30], as in this specimen. A similar eggshell structure, with a thick organic layer (over 100 μm) and a very thin mineral layer (less than 10 μm), has also been documented in isolated eggs of the basal choristoderan *Hyphalosaurus baitaigouensis*[31]. These shell structures reveal that the aquatic choristoderes were oviparous and presumably came ashore to lay their eggs, like extant sea turtles and crocodilians. Putative females of the neochoristoderan *Champsosaurus* possessed fused sacral centra and more robust limb bones than males, perhaps adaptations for nesting on land[32]. Other choristoderes were viviparous, such as one specimen of *H. baitaigouensis*[28] and *Monjurosuchus splendens*[14]. The co-occurrence of viviparity and oviparity in one nominal species, *H. baitaigouensis*, indicates that it had a bimodal reproductive mode, that is, both viviparity and oviparity occurred in a single species as in the basalmost sauropsid *Mesosaurus tenuidens* and few extant squamates[14,33].

## Macroevolutionary study

The new specimen exemplifies the complexity of parity modes in some early reptiles and provides information to constrain the evolution of reproductive strategies in archosauromorphs, reptiles and amniotes in general. We return to the question of whether EER is the ancestral condition. To test this hypothesis, we collected data on reproductive modes from representative taxa spanning the phylogenetic diversity of extant amniotes, and from all extinct taxa where information is available (Supplementary Table 1). The balance of data on fossil taxa inevitably favours those that laid rigid-shelled eggs because those are preserved more readily than eggs without mineralized shells. In particular, we could find no examples of extinct synapsids with evidence of reproductive mode. However, we compensated for this by broadly sampling extant taxa for which reproductive data are secure. Most oviparous squamates lay membrane- and parchment-shelled eggs at the limb-bud stage. In contrast, most amniotes that lay rigid-shelled eggs obligately oviposit at an early developmental stage, for instance, at the blastula stage in birds, the gastrula stage in chelonians and tuataras, and the neurula stage in crocodilians[33–35]. In these forms, the thick calcite layer delays the exchange rate of respiratory gases and may prevent the development of the embryo before the eggs are laid[36]. Therefore, both eggshell and developmental stage of the embryo at oviposition provide information constraining the ancestral state of reproductive modes.

We coded each extant and extinct taxon for three characters: (1) reproduction mode: viviparous, oviparous; (2) eggshell mineralization: membrane-shelled, parchment, rigid; (3) EER: absent, present. EER was defined as amniotes that lay eggs at the limb-bud stage or later[33]. EER was identified in extinct taxa based on fossil adults that contain embryos at the limb-bud stage or later or are associated with neonates, which were commonly identified as evidence for viviparity previously (Supplementary Table 1). We conservatively treated all fossils for which we could not judge the stage of development of an egg at oviposition as non-EER. Character 2 is dependent on character 1, so viviparous taxa were coded as inapplicable ('−') for character 2. To resolve the problem of character dependency, characters 1 and 2 were amalgamated into a single structured Markov model (SMM) equipped with hidden states. We also applied an analogous approach within a parsimony framework using Sankoff (cost) matrices. We conducted an exhaustive multiplicative set of ancestral states analyses, accounting for: (1) different

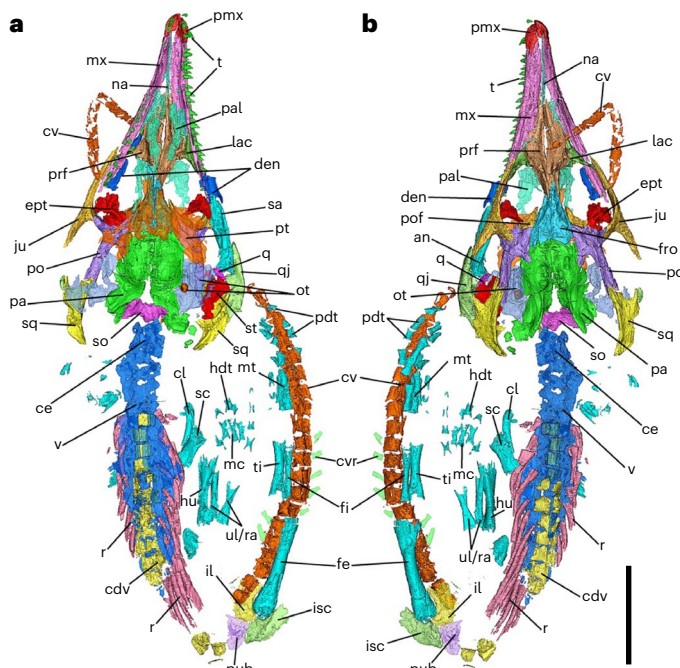

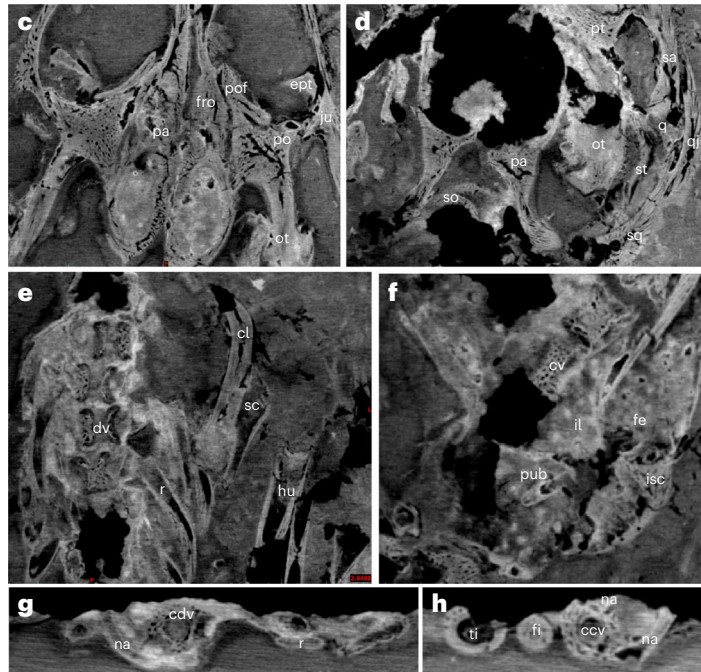

**Fig. 3 | CT scans of the choristodere skeleton (*Ikechosaurus* sp.) and its reconstruction. a,b**, Ventral (**a**) and dorsal (**b**) views of the reconstructed skeleton. **c–f**, Coronal slices showing dorsal (**c**) and ventral (**d**) sections of the posterior skull, proximal left forelimb and presacral axial bones (**e**), and pelvic girdle, proximal axial bones and femur (**f**). **g,h**, Transverse slices show the relationship of centra and neural arches in dorsal (**g**) and caudal (**h**) vertebrae. cdv, centra of dorsal vertebrae; ce, cervical; cl, clavicle; cv, caudal vertebra; den,

dentary; ept, ectopterygoid; fe, femur; fi, fibula; fro, frontal; hdt, hand digit; hu, humerus; il, ilium; isc, ischium; ju, jugal; lac, lacrimal; mc, metacarpal; mt, metatarsal; mx, maxillary; na, nasal; ot, otic area; pa, parietal; pal, palatine; pdt, pes digit; pmx, premaxillary; po, postorbital; pof, postfrontal; prf, prefrontal; pt, pterygoid; pub, pubis; q, quadrate; qj, quadratojugal; r, rib; ra, radius; sa, surangular; sc, scapula; so, supraoccipital; sq, squamosal; st, stapes; t, tooth; ti, tibia; ul, ulna; v, vertebra. Scale bar, 10 mm.

phylogenetic time-scaling methods; (2) alternative tree topologies; (3) exclusion of key fossils; (4) different ancestral state reconstruction methods, evolutionary models and optimization criteria; (5) among lineage rate heterogeneity; and (6) constraining extant node states based on previous studies. Together, these totalled over 100,000 individual analyses. Further details can be found in the Methods.

The ancestral state reconstruction results are unequivocal for both the amalgamated character (reproduction mode + eggshell mineralization) and EER presence (Fig. 4). Viviparity with EER dominates the deeper nodes, being the most likely condition for the roots of Amniota (mean marginal maximum likelihood (ML)-based ancestral state across 100 trees for the best-fitting model and other models whose Akaike information criterion (AIC) difference was less than 2 compared to the best-fitting model for viviparity-ML: 99.5–100%; EER-ML: 99.8–100%), Reptilia (viviparity-ML: 98.5–100%; EER-ML: 100%), Diapsida (sensu lato) (viviparity-ML: 100%; EER-ML: 100%), Archelosauria (viviparity-ML: 98.5–100%; EER-ML: 99.6–100%) and Archosauromorpha (viviparity-ML: 99.4–100%; EER-ML: 99.8–100%), irrespective of which time-scaling approach or best-fitting model (including models whose AIC differed from the best-fitting model by less than 2) was considered (Supplementary Tables 3–6).

Using a different ancestral state reconstruction also does not change the main conclusions: 10 of 18 maximum parsimony (MP)-based ancestral state reconstructions recover viviparity at the origin of Amniota, with the remaining reconstructions mostly favouring membrane-shelled eggs as the ancestral state of amniotes (all MP-based reconstructions using accelerated transformation (ACCTRAN) favour viviparity at the origin of Amniota; Supplementary Tables 21–26). All MP-based ancestral state reconstructions favour EER as the ancestral state of Amniota. Bayesian traits (BT) results are consistent with ML-based ancestral state reconstructions (Supplementary Tables 28–31 and Extended Data Fig. 4). Fixing the nodes of both

Lepidosauria and Squamata to a non-viviparous state (Supplementary Tables 28–31 and Extended Data Fig. 5) still results in viviparity being the most likely condition for the origin of Amniota (mean BT-based ancestral state for viviparity: 87.4–97.7%), only leading to increased variability in the ancestral state estimates (Reptilia viviparity-BT: 79.8–97.2%; Diapsida (sensu lato) viviparity-BT: 72.3–99.7%) and potentially suggesting that viviparity re-evolved in Archosauromorpha (Archelosauria viviparity-BT: 16.6–91.2%; Archosauromorpha viviparity-BT: 48.9–96.1%).

Among archosaurs, the results reflect the possession of rigid-shelled eggs by extant birds and crocodilians, which is consistent with the proposal[15] that the first dinosaurian eggs were membrane-shelled, although the results are less clear. Most best-fitting models favour rigid-shelled eggs at the root of Theropoda (rigid-ML: 81.6–98.5%; EER-ML: 0–1.5%); however, non-EER, membrane-shelled eggs are most likely at the root of Saurischia (membrane-shelled-ML: 67.5–97.2%; EER-ML: 0.1–0.4%), Dinosauria (membrane-shelled-ML: 60.5–96.7%; EER-ML: 0.1–1.4%) and probably also Archosauria (membrane-shelled-ML: 52–96.1%; EER-ML: 0.2–13.6%). The non-EER results are consistent across all variant analyses but the reproduction mode and eggshell mineralization results are somewhat equivocal for later-diverging clades. The results from the best-fitting models are compatible with parchment-shelled eggs as the ancestral state for Saurischia (parchment-ML: 1–30.3%), Dinosauria (parchment-ML: 1.1–35.6%) and Archosauria (parchment-ML: 1.3–42.5%). If we also consider simpler ML models with a worse fit (AIC difference greater than 2), where character state transition rates are more constrained compared to the best-fitting models, the evidence for parchment-shelled eggs for these clades increases. Note, however, that parsimony-based analyses favour rigid-shelled eggs as the ancestral state of the three clades. Bayesian analyses are generally consistent with ML-based results but evidence for parchment-shelled eggs in

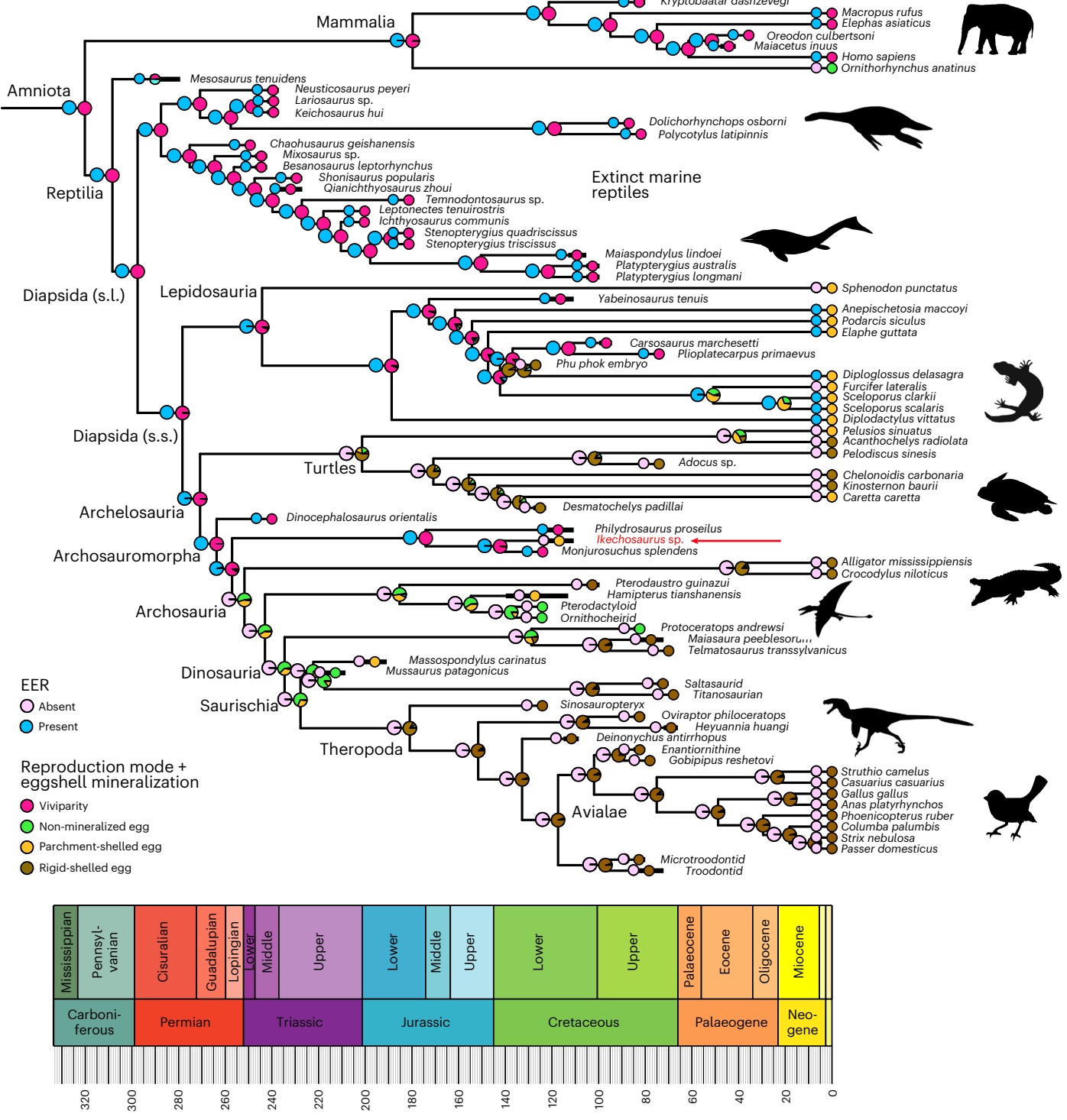

**Fig. 4 | Phylogeny of amniotes, showing known reproduction mode and eggshell mineralization, and the EER of 80 extant and extinct species (tips, to right), and the inferred mean ancestral states for all branching points (larger pie charts at the nodes).** The dominant inferred state towards the root (left) is viviparity with EER. This is a consensus tree based on a sample of 100 trees time-scaled using the fossilized birth-death (FBD) tip-dating method (with node age constraints for major clades) and component ARD model with a switch-on dependency (CARD_sw) for eggshell mineralization and reproduction mode (best-fitting ML model based on the AIC score; Supplementary Table 6). Note that alternative evolutionary models do not recover viviparity as the ancestral state in Lepidosauria (Supplementary Tables 3–6). *Ikechosaurus* sp. is shown in red and indicated by a red arrow. Elephant and bird icons reproduced from PhyloPic under a Creative Commons license CC BY 3.0 (elephant, T. Michael Keesey; bird, Chloé Schmidt).

these three clades is generally lower; fixing the node of Lepidosauria and Squamata to a non-viviparous state leads to a higher variability in the ancestral state estimates. These discrepancies emphasize that while the ancestral state results are consistent for deep nodes, the results for later-diverging clades should be treated with more caution. A similar example is found in Lepidosauria for which parsimony and alternative ML and BT evolutionary models do not recover viviparity as ancestral (Supplementary Tables 3–26 and 28–45).

While the best EER model is consistently recovered as all-rates-different (ARD), the best model to explain character state evolution for reproduction mode and eggshell mineralization is more enigmatic and changes depending on the distribution of branch lengths, the time-scaling methods used and topology (Supplementary Tables 27 and 28–45). However, the component ARD model with or without a switch-on dependency is always among the best-fitting ML-based models irrespective of which time-scaling approach or topology is considered (Supplementary Tables 3–20). The component ARD model is generally also found among the best-fitting BT-based models; however, the evidence for alternative model hypotheses is generally lower for analyses with fossilized Lepidosauria and Squamata node states due to smaller-log Bayesian factors (BFs) (Supplementary Tables 28–45).

## Discussion

It is widely accepted that viviparity evolved from oviparity through EER in squamates and mammals[33–35], and this may also have been the case in the various aquatic vertebrates of the past, such as choristoderes, as well as mesosaurs, ichthyosaurs and some sauropterygians. This implies that viviparity might have derived from membrane- or parchment-shelled eggs with EER in these clades. Yet, our results strongly suggest viviparity as the ancestral condition of amniotes and the marine reptile clades.

This result is subject to a number of biases: first, the phylogenetic placement of extinct marine reptiles within the Amniota is contested[37]. Our results, however, are qualitatively unchanged if we revise the phylogeny to place extinct marine reptiles within the Lepidosauromorpha or in other basal diapsid locations (Supplementary Tables 15–20 and 40–45). Completely removing the extinct marine reptiles from the analyses has little impact on the results, with only models with a worse fit and some of the models for which the states of both Lepidosauria and Squamata were fixed a priori showing more uncertainty in the ancestral state reconstructions (Supplementary Table 11–14 and 36–39). Further, the placement of *Mesosaurus*, coded ambiguously as either viviparous or oviparous with non-mineralized eggs[14,17] near the root of the Amniota does not affect these results; if it is removed, the results remain qualitatively the same (Supplementary Tables 7–10 and 32–35).

Second, our dataset may be taphonomically biased. It could be argued that well-skeletonized embryos in adult oviducts are more likely to be preserved than membrane- or parchment-shelled eggs, and that isolated eggs are difficult to assign to species. Consequently, viviparity might be overrepresented by extinct viviparous clades in our analysis. Given that *Ikechosaurus* is a derived choristodere[22], the discovery of oviparity in this presumably viviparous clade suggests that either oviparity evolved from viviparity or viviparity evolved from extinct oviparous ancestors with EER in choristoderes (the latter hypothesis, however, is not backed by the fossil record). This might also be true for extinct viviparous clades, such as mesosaurs, ichthyosaurs and sauropterygians that belong to basal parareptiles or amniotes in general and diapsids or lepidomorphs, respectively. As fully aquatic amniotes could not come onto land to lay their eggs and embryos in eggs could not survive in water, viviparity in these extinct amniotes must have evolved from either semi-aquatic or terrestrial viviparous ancestors (as implicated by our analyses), or oviparous ancestors that displayed EER but whose eggs have not been preserved in the fossil record[27]. In either case, these early amniotes reflect EER as the primitive reproductive mode of amniotes (Fig. 1, model on the right).

The absence of rigid-shelled eggs through the Carboniferous period, Permian period and most of the Triassic period[2,14,15] has long been noted: if such eggs truly existed widely during this time interval, it could change our results entirely, pointing to rigid-shelled eggs as the ancestral amniote condition. However, as rigid-shelled eggs are more likely to be preserved than membrane- or parchment-shelled eggs and perhaps the partially ossified and tiny bones of retained

embryos within the mother, this might not be a preservation bias but a real absence[15]. Conversely, given the scarcity of fossil evidence on the reproductive mode of Palaeozoic amniotes and the low fossilization potential of membrane- or parchment-shelled eggs, the possibility remains that the earliest amniote eggs were, indeed, membrane- or parchment-shelled. The addition of more extinct taxa will provide an important test of these results[14,15].

The occurrence of parchment-shelled eggs associated with EER in choristoderes extends the wide occurrence of this phenomenon among amniotes and may be the primitive reproductive mode that occurred before archosaurs (crocodilians, dinosaurs, birds) and chelonians acquired non-EER oviparity and rigid-shelled eggs[1]. As membrane- and parchment-shelled eggs associated with EER are also common in extant mammals and squamates, this suggests that this condition was common among early terrestrial amniotes, for example, in the first 100 Myr of their evolution from the Carboniferous to the Triassic, favouring a model that amniotic fetal membranes evolved in association with EER. Similarly, a phylogenetic study of extant tetrapods[38] inferred that many of the structures that characterize the Amniota (delayed deposition of eggs, large yolk mass, cellular yolk sac and amnion) may have evolved in an aquatic environment in association with delayed egg laying.

## Methods

### CT scanning and reconstruction

The specimen was scanned at the University of Bristol, UK, on the Nikon X-Tek H 225 ST X-ray scanner at 225 kV and 188 µA (42.3 W) from a rotating tungsten target, with 2-s exposure, 1× binning, 24-dB gain and a 3-mm copper filter, slice thickness = 48.45 µm and total number of slices = 1,142. Each scan captured 3,141 projections, with four frames averaged per projection. The reconstructed scan data were subsequently combined in VGStudio v.3 (https://www.volumegraphics.com). A three-dimensional (3D) model was created from the CT data using the segmentation tools in Avizo v.9.1.1 Lite (Visualization Science Group; https://www.fei.com/software/amira-avizo/). All scan data and 3D models are available in the supplementary data.

### Scanning electron microscopy

Samples were examined using a JEOL 8530F Hyperprobe at the School of Earth Sciences, University of Bristol, and a LEO 1530VP scanning electron microscope at the Technical Services Centre, Nanjing Institute of Geology and Palaeontology, Chinese Academy of Sciences. Both instruments were equipped with a secondary electron detector, a back-scattered electron detector and an energy dispersive X-ray spectrometer.

### Phylogenetic macroevolutionary analysis

**Data collation.** Data were compiled on key reproductive parameters for as many extinct amniotes as possible, distinguishing three characters, each with two or three states: reproduction mode: (1) viviparity and (2) oviparity; eggshell mineralization: (1) non-mineralized, (2) weakly mineralized, (3) rigid; and EER: (1) absent and (2) present. These eggshell and parity characteristics have been documented widely in the literature, and we indicate exact data sources in our data compilation (Supplementary Table 1). We identified 59 extinct taxa for which eggs or viviparity had been identified (6 mammals; 1 mesosaur; 6 turtles; 13 ichthyosaurs; 5 sauropterygians; 4 squamates; 3 choristoderes; 1 protorosaur; 2 crocodilians; 4 pterosaurs; 13 dinosaurs; 1 bird). We added a further 21 extant taxa, making a total of 80. As the basis for an initial analysis of ancestral states for Amniota and subclades, we compiled a supertree for the 80 taxa, using a standard genomic tree[39] as scaffold (Supplementary Table 2), supplemented by recent cladistic analyses of extinct groups[40]. *Seymouria baylorensis* and *Diadectes sideropelicus* were added as outgroups to a polytomy including the Amniota.

**Phylogenetic time-scaling.** We obtained first and last appearance dates (FAD and LAD, respectively) for each taxon in our analysis using the Paleobiology Database[41]. We then time-scaled the supertree using four different approaches: (1) the minimum branch length (mbl) method[42]; (2) the equal branch length (equal) method[43]; (3) the FBD tip-dating method[44–46] with only the root; and (4) with both the root and the node ages of major extant clades constrained. The mbl and equal methods are a posteriori time-scaling methods[47] that avoid zero-length branches by imposing a minimum branch length of 1 Myr (mbl) or by taking an equal share from preceding non-zero-length branches (equal). We generated 100 time-scaled trees with the time-PaleoPhy function of the paleotree package[48] with tip dates sampled from a uniform distribution (dateTreatment = minMax) bounded by FADs and LADs and the vartime argument set to 1 Myr. The FBD method jointly considers speciation, extinction and fossil preservation rates to estimate divergence times in a Bayesian framework[44–46]. We applied a clockless tip-dating approach using MrBayes (v.3.2.7a)[49,50] where the empty morphological matrix was generated using the createMrBayes-TipDatingNexus function of the paleotree package[48]. Topology was constrained to the input supertree; uniform priors bounded by the respective FADs and LADs were used to calibrate the tip ages. To place the time-scaled trees in absolute time, *M. tenuidens* (Early Permian, Kungurian, 278.4 Myr) was used as the anchor taxon. For all node calibrations, we used offset gamma distributions with a shape parameter of 3. Based on Benton et al.[51] for the root of the Amniota, we used an offset gamma distribution with a mean age of 325.1 Myr, a minimum age of 318 Myr and an s.d. of 4.099175 Myr. Additional major crown clade calibrations[50] for the second FBD analysis were parametrized as follows: Mammalia (minimum age = 164.9, mean age = 182.3402, s.d. = 10.06911); Diapsida (255.9, 274.9603, 11.0045); Lepidosauria (238, 245.0047, 4.044152); Squamata (168.9, 188.2463, 11.16956); Archosauria (247.1, 253.3423, 3.603972); and Aves (66, 75.91138, 5.722338). The default FBD and clock priors[52] provided by createMrBayesTipDatingNexus were kept. We disallowed sampled ancestors (prset samplestrat = fossiltip;). we ran the two FBD analyses four times, using four chains per run, for 1,000,000,000 generations, sampling every 100,000th generation. We checked convergence using Tracer v.1.7.1 (ref. [53]), ascertaining an effective sample size (ESS) of all parameters exceeding 200 for combined traces. We used the obtainDatedPosteriorTreesMrB function of the paleotree package to obtain a sample of 100 time-scaled trees from the posterior, employing a burn-in of 50%. Before the ancestral state reconstruction, we removed the outgroup taxa *S. baylorensis* and *D. sideropelicus*.

**Ancestral state estimation.** Hierarchical character dependencies have long presented a challenge in ancestral state estimation[54–56]. In our analysis, character 1 and 2 are hierarchically related; the state of character 2 (eggshell mineralization) is dependent on character 1 (reproduction mode) being in a specific state (state = oviparous). This is equivalent to the classic tail colour problem of Maddison[54]. Recently, Tarasov[55] demonstrated that SMMs equipped with hidden states solve the problem of modelling character complexes with hierarchical dependencies. In doing so, he also demonstrated the invariant nature of characters and states. These concepts are equivalent; characters can be transformed into states and vice versa. We followed the approach of Tarasov[55] and amalgamated characters 1 and 2 into a single SMM with 6 states (Extended Data Fig. 3). Ancestral states were estimated under two variations of this model. SMM_ind assumes that reproduction mode and eggshell mineralization evolve independently. This is analogous to modelling the characters separately using two independent models, except for the fact that under the SMM approach, simultaneous changes to character 1 and character 2 are prohibited. Alternatively, SMM_sw assumes 'switch-on' dependency. That is, character 2 (eggshell mineralization) can only change state if character 1 (reproduction mode) is in a specific state (state = oviparous).

The SMM_ind and SMM_sw approaches consider different models on rate transition between the character states, resulting in eight evolutionary models: a component equal-rate (ER) model (CER_ind and CER_sw: transitions between states among component characters share a single rate parameter); a component symmetrical model (CSYM_ind and CSYM_sw: transitions between states among component characters are symmetrical); a component ARD model (CARD_ind and CARD_sw: transitions between states among component characters are all different); and an equal rates model (ER_ind and ER_sw: transitions between aggregated rates share a single rate parameter). The SMMs used in this study are summarized in Extended Data Fig. 3.

We used an ML approach to estimate the ancestral states for the SMMs, applying the asr_mk_model function (with the optimization algorithm set to 'optim') of the castor R package[57]. To avoid optimization problems, the input trees were scaled to a tree height of 1 before the ML analyses. The transition matrix was fitted ten times and the maximum allowed number of iterations per fitting trial was set to 500. We used the 'tip.priors' argument to assign probabilities to amalgamated states. *M. tenuidens* is a special case. Current fossil evidence does not enable us to determine confidently whether *M. tenuidens* was viviparous or oviparous with membrane-shelled eggs[14,17]. Again, we used the tip.priors argument to specify this uncertainty. For the CARD models, which are not time-reversible, marginal ancestral likelihoods were computed without rerooting the input tree. We used the AIC to select the best-fitting model. We then calculated the mean marginal ancestral states of the best-fitting model for each set of 100 input trees, which were plotted on a consensus tree generated using the consensus. edges function of the R package phytools[58]. Plots were generated using the R package strap[59].

The presence and absence of EER was also modelled using asr_mk_ model with default parameter settings, the optimization algorithm set to optim, using the same input trees, and providing the two-state character for EER via the 'tip_states' argument. We used an ER model (EER ER) and an ARD model (EER ARD). Model selection was again carried out using the AIC. The calculation and plotting of mean marginal ancestral states followed the same practice as for the SMMs.

In addition to ML, we also ran an MP-based ancestral state reconstruction for the same characters using the ancestral.pars function of the phangorn package[60,61]. As the parsimony approach does not estimate transition rates, the evolutionary models used in the ML approach cannot fully be translated into a parsimony setting. We generated three parsimony models for the amalgamated character (reproductive mode and mineralization): one ACCTRAN approach that allows all character transitions and two most-parsimonious reconstructions (MPR) approaches that attempt to model either independence of reproduction mode and eggshell mineralization (similar to SMM_ind) or a switch-on dependency (similar to SMM_sw) using Sankoff (cost) matrices[54,56]. The number of required steps for transitions that were forbidden in the model was set to a value (step cost = 100) that would make it practically impossible for the transition to occur. The ACCTRAN[62] approach as implemented by ancestral.pars does not allow for cost matrices, thus requiring the MPR[63,64] approach. Contrast matrices were used to assign probabilities to amalgamated states and account for uncertainty in tip states. The MP-based ancestral state reconstruction was repeated for the EER character using ACCTRAN.

Furthermore, we also ran a BT ancestral state reconstruction using the package BayesTraits[65,66] for each set of 100 time-scaled input trees. As with the ML and MP approaches, the input data was formatted to account for uncertainty in (amalgamated) tip states. To avoid optimization problems, input time-scaled trees were rescaled to have a mean branch length of 0.01. We used the same SMM and EER used in the ML approach. A reverse-jump, continuous time Markov[67,68] Chain Monte Carlo algorithm (rjMCMC) was applied to both homogeneous and variable rate models, the latter allowing for shifts in the rate of evolution $\sigma_v^2$ on individual branches[69,70]. For the models with multiple

transition rate parameters, we ran the multistate approach using the rjMCMC method with an exponential (0, 10) hyperprior. For the single-rate models (ER_ind, ER_sw, EER ER), a uniform (0, 10) distribution was set as the transition rate prior. Three independent MCMC chains per model were run for 11,000,000 iterations and parameters were sampled every 10,000 iterations; 1,000,000 iterations were discarded as burn-in. We calculated the ML of the models using the stepping stone sampler[71] implemented in BayesTraits. We sampled 1,000 stones and used 100,000 iterations per stone. Convergence was assessed using the R package CODA[72], ensuring that the smallest ESS always exceeded 200 for the combined chains. Models were compared using a log BF test[73] applied to the mean log MLs from the combined three MCMC chains. To calculate the log BF, the homogeneous rate ER_sw and homogeneous rate EER ER models served as the simple comparison models for the SMMs and the EER models, respectively. Mean ancestral states were calculated across the combined three MCMC chains for each model and plotting followed the same practice used for the ML approach. Given the uncertainty in the ancestral state reconstructions for Lepidosauria and Squamata[8-10,74-77], we reran our Bayesian SMM analyses, this time fixing the nodes of the two clades to a non-viviparous state.

**Robustness tests.** To test the robustness of our results, we reran all our analyses dropping *M. tenuidens* and extinct marine reptiles, respectively, from our input phylogeny. We also reran the mbl and equal time-scaled trees with a modified phylogenetic position of the extinct marine reptiles, adding them either as a sister taxon to Archelosauria, Archosauromorpha or Lepidosauria[21,78,79].

### Reporting summary

Further information on research design is available in the Nature Portfolio Reporting Summary linked to this article.

## Data availability

We provide all data as supplementary data. The phylogeny used in this study is shown in Fig. 4. The specimen studied (MES-NJU 57004) is hosted at the School of Earth Sciences and Engineering, Nanjing University. Correspondence and requests for materials should be addressed to B.J. or M.J.B.

## Code availability

All analyses in this study were conducted using readily available, published programs that are cited in the text. The versions of the programs are as follows: R v.4.1.0; ape v.5.5; castor v.1.6.7; paleotree v.3.3.25; phangorn v.2.7.0; phytools v.0.7-70; strap v.1.4; and BayesTraits v.4.0.0.

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

## Acknowledgements

We thank C. O'Donovan, M. Pagel, G. Ruxton and M. Sakamoto for discussions during the course of this study, and M. Laurin and G. Wagner for comments. B.J. was supported by the National Science Foundation of China (award no. 42288201), Strategic Priority Research Program (B) of the Chinese Academy of Sciences (award no. XDB26000000) and Fundamental Research Funds for the Central Universities (award no. 0206-14380137). M.J.B. was funded by a Natural Environment Research Council (NERC) UK (grant no. NE/P013724/1) and European Research Council Advanced Grant (no. 788203). A.E. was funded by NERC UK grant nos. NE/L002434/1 and NE/P013724/1. This work was carried out using the computational facilities of the Advanced Computing Research Centre, University of Bristol (http://www.bris.ac.uk/acrc/).

## Author contributions

B.J. and M.J.B. conceived the study. Y.H. and B.J. made the 3D CT reconstruction. S.W. and S.L.K. carried out the scanning electron microscopy and geochemical analyses of the specimen. A.E., J.N.K., J.S. and M.J.B. carried out the phylogenetic comparative analyses. All authors contributed to data collection, interpreted the results and wrote the manuscript.

## Competing interests

The authors declare no competing interests.

## Additional information

**Extended data** is available for this paper at https://doi.org/10.1038/s41559-023-02074-0.

**Correspondence and requests for materials** should be addressed to Baoyu Jiang.

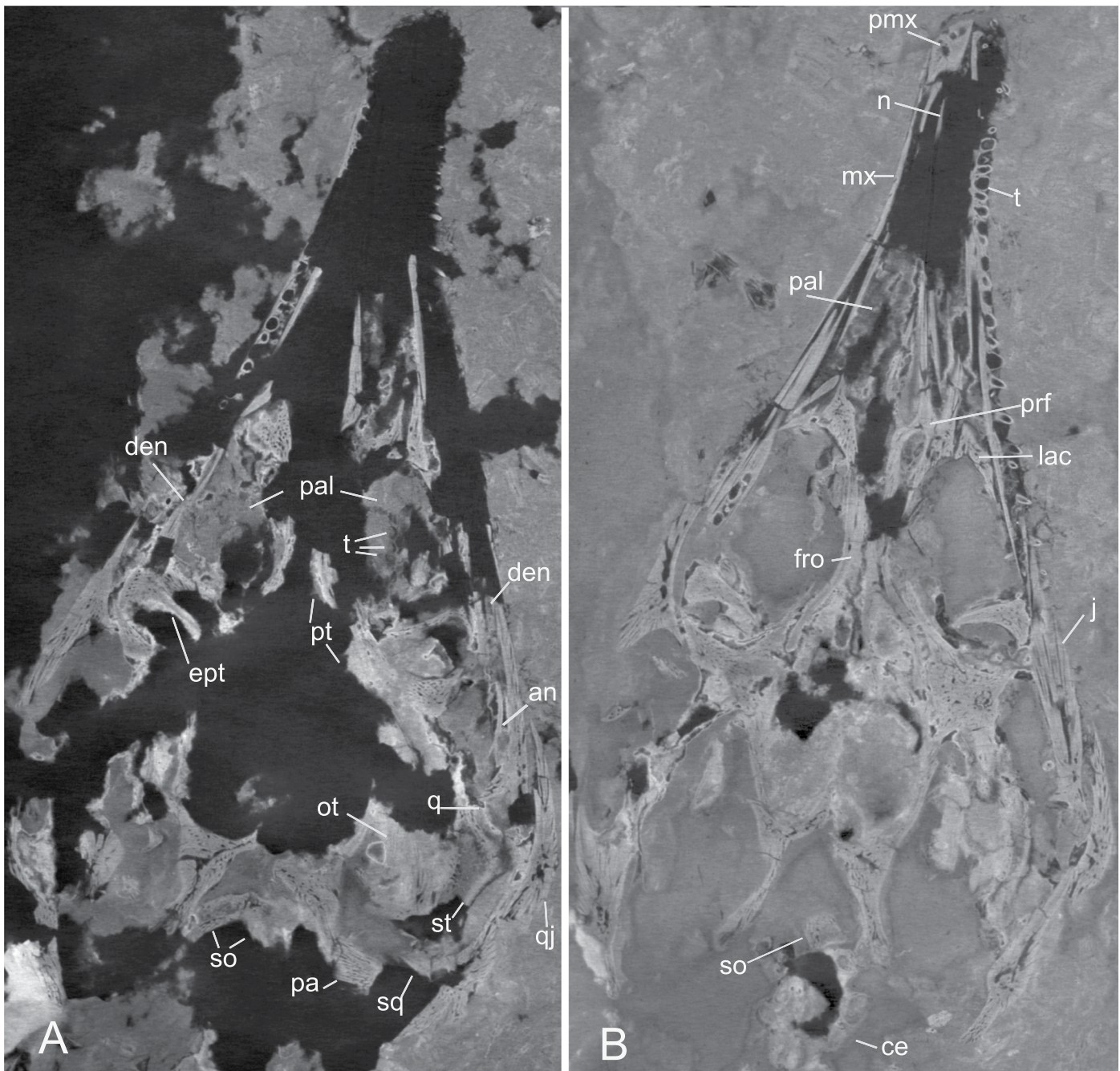

**Extended Data Fig. 1 | Coronal CT slices of the embryo skull. a**. Approximate central view. **b**. Approximate middle views. Abbreviations see Fig. 1.

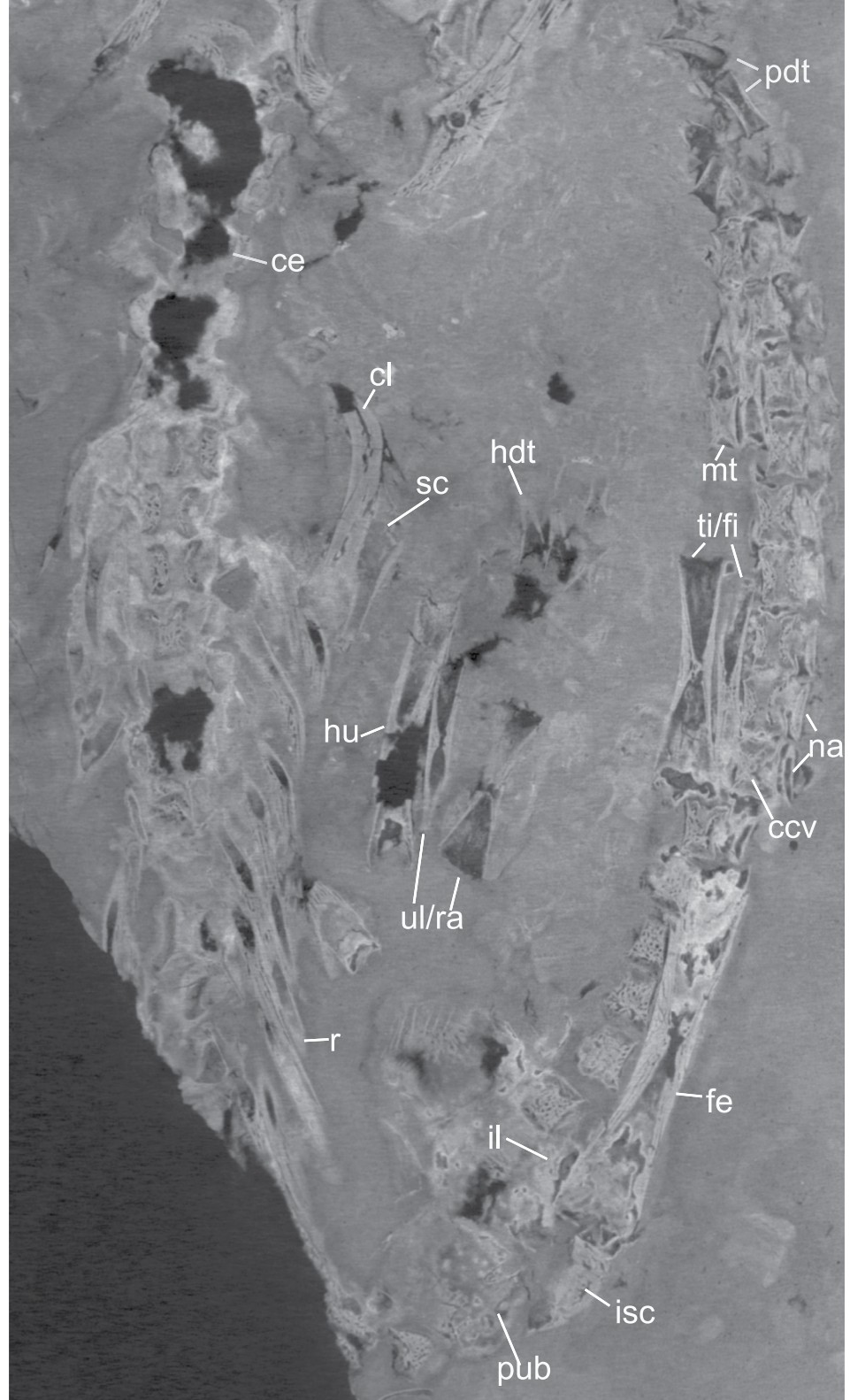

**Extended Data Fig. 2 | A coronal CT slice shows axial and appendicular skeletons of the embryo.** Abbreviations see Fig. 1.

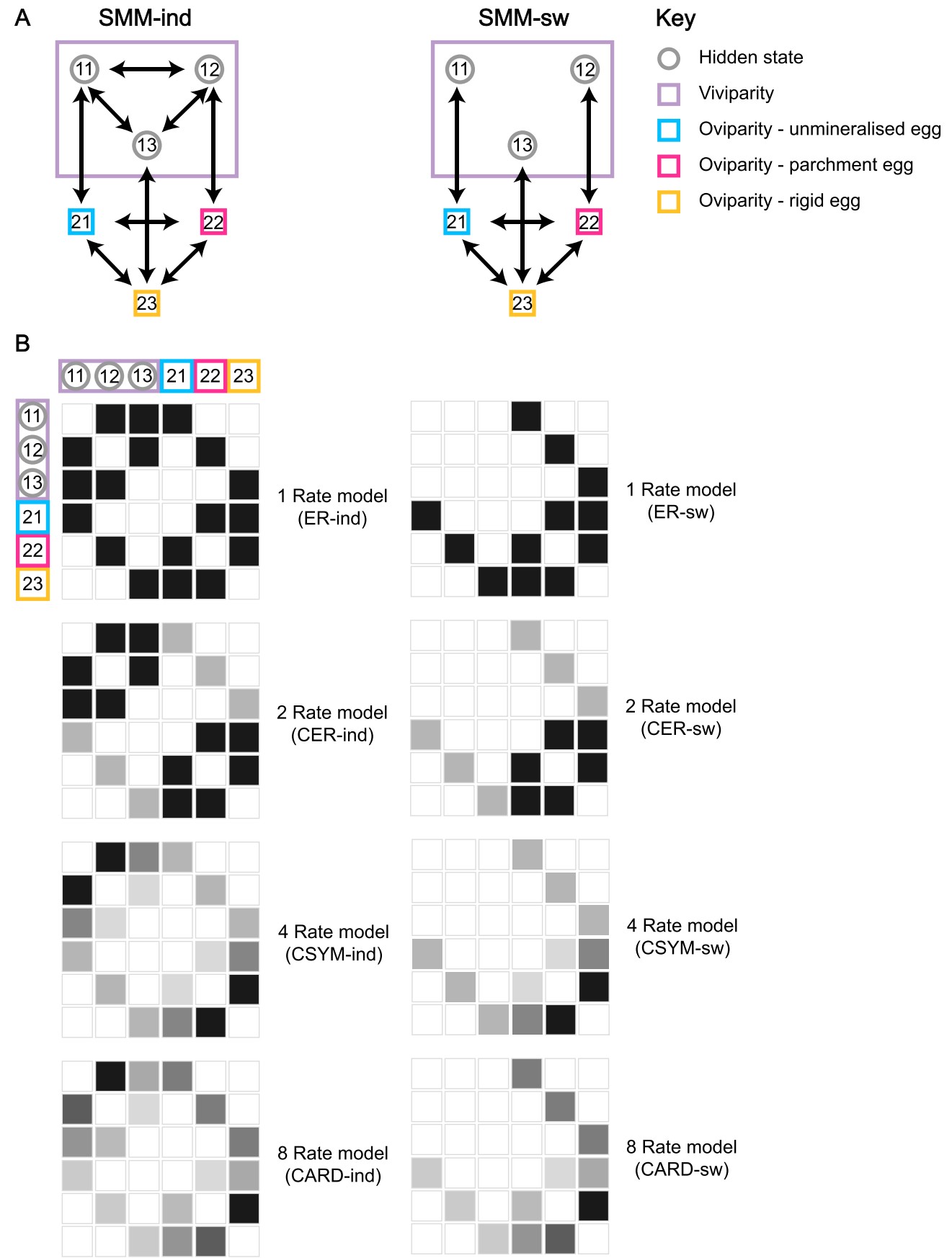

**Extended Data Fig. 3 | See next page for caption.**

**Extended Data Fig. 3 | Visualisation of the Structured Markov Models (SMMs) used in this study. a.** Diagrammatic representation of the SMMs. SMM-ind = independent model; SMM-sw = switch-on dependency model. Numbers represent amalgamated character states. Left digit represents viviparity = 1, oviparity = 2. Right digit represents unmineralized egg = 1, parchment egg = 2, rigid egg = 3. **b**. Index matrices for the eight SMM's used in the ancestral state analyses. Different greyscale values represent different rate categories. Equal rates (ER) models: transitions between aggregated rates share a single rate parameter; Component equal-rates (CER) models: transitions between states among component characters share a single rate parameter; Component symmetrical (CSYM) models: transitions between states among component characters are symmetrical; Component all-rates-different (CARD) models: transitions between states among component characters are all different.

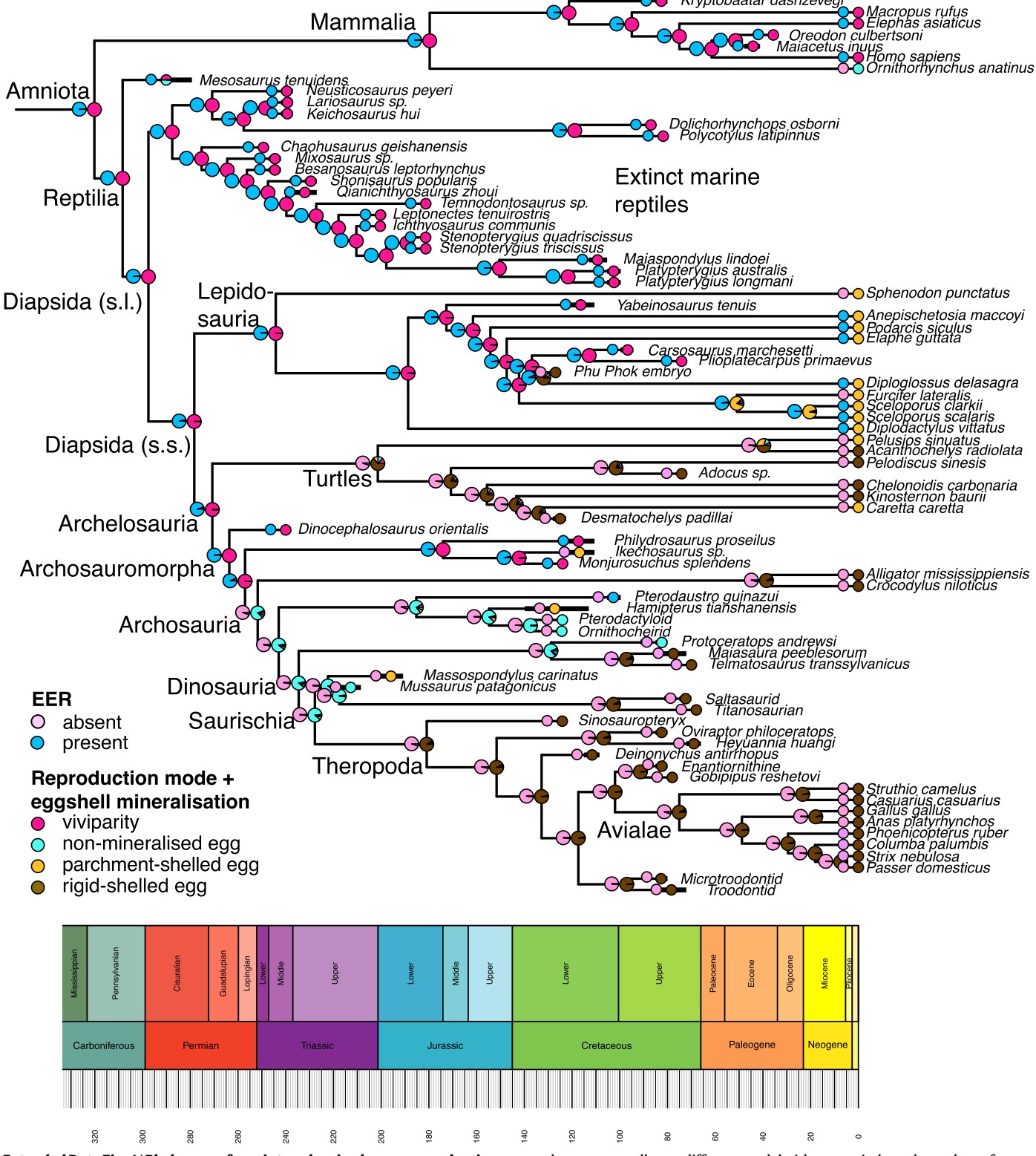

**Extended Data Fig. 4 | Phylogeny of amniotes, showing known reproduction mode + eggshell mineralisation, and EER of 80 extant and extinct species (tips, to right), and inferred mean ancestral states for all branching points (larger pie charts at nodes).** The dominant inferred state towards the root (left) is viviparity with EER. This is a consensus tree based on a sample of 100 trees time-scaled using the FBD method (with node age constraints for major clades) and component all rates different model without a switch-on dependency for eggshell mineralisation and reproduction mode (CARD_ind.het) (best-fitting BT model based on log BF score; see Supplementary Table 31a–d), which allows for variable evolutionary rates on individual branches. Ikechosaurus sp. is indicated in red and with a red arrow. Silhouettes as in Fig. 4.

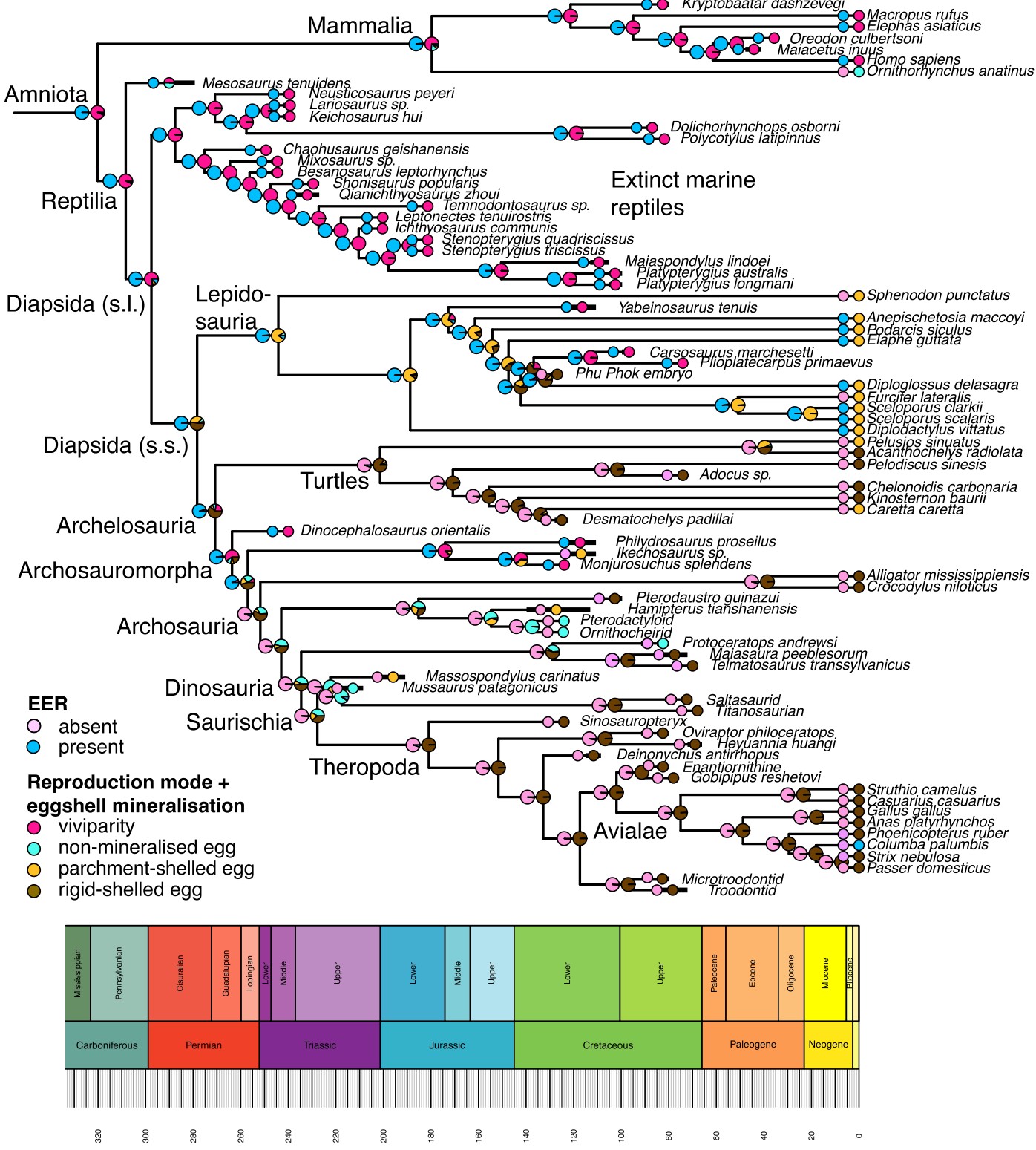

**Extended Data Fig. 5 | Same as Supplementary Fig. 4, but the nodes of Lepidosauria and Squamata have been constrained to a non-viviparous state.** The best-fitting BT model for eggshell mineralisation and reproduction mode presented here is a component equal-rates model without a switch-on dependency for eggshell mineralisation and reproduction mode (CER_ind.het) (see Supplementary Table 31d–g), which allows for variable evolutionary rates on individual branches.

# Reporting Summary

## Statistics

For all statistical analyses, confirm that the following items are present in the figure legend, table legend, main text, or Methods section.

| n/a | Confirmed | |
|---|---|---|
| ☐ | ☒ | The exact sample size (*n*) for each experimental group/condition, given as a discrete number and unit of measurement |
| ☒ | ☐ | A statement on whether measurements were taken from distinct samples or whether the same sample was measured repeatedly |
| ☒ | ☐ | The statistical test(s) used AND whether they are one- or two-sided *Only common tests should be described solely by name; describe more complex techniques in the Methods section.* |
| ☒ | ☐ | A description of all covariates tested |
| ☒ | ☐ | A description of any assumptions or corrections, such as tests of normality and adjustment for multiple comparisons |
| ☒ | ☐ | A full description of the statistical parameters including central tendency (e.g. means) or other basic estimates (e.g. regression coefficient) AND variation (e.g. standard deviation) or associated estimates of uncertainty (e.g. confidence intervals) |
| ☒ | ☐ | For null hypothesis testing, the test statistic (e.g. *F*, *t*, *r*) with confidence intervals, effect sizes, degrees of freedom and *P* value noted *Give P values as exact values whenever suitable.* |
| ☐ | ☒ | For Bayesian analysis, information on the choice of priors and Markov chain Monte Carlo settings |
| ☐ | ☒ | For hierarchical and complex designs, identification of the appropriate level for tests and full reporting of outcomes |
| ☒ | ☐ | Estimates of effect sizes (e.g. Cohen's *d*, Pearson's *r*), indicating how they were calculated |

*Our web collection on statistics for biologists contains articles on many of the points above.*

## Software and code

Policy information about availability of computer code

| Data collection | *Provide a description of all commercial, open source and custom code used to collect the data in this study, specifying the version used OR state that no software was used.* |
|---|---|
| Data analysis | All analyses in this study were conducted using readily available, published programs and are cited in the text. Version numbers of the programs we used are as follows: R version 4.1.0; ape 5.5; castor 1.6.7; paleotree 3.3.25; phangorn 2.7.0; phytools 0.7-70; strap 1.4. |

For manuscripts utilizing custom algorithms or software that are central to the research but not yet described in published literature, software must be made available to editors and reviewers. We strongly encourage code deposition in a community repository (e.g. GitHub). See the Nature Portfolio guidelines for submitting code & software for further information.

## Data

Policy information about availability of data

All manuscripts must include a data availability statement. This statement should provide the following information, where applicable:
- Accession codes, unique identifiers, or web links for publicly available datasets
- A description of any restrictions on data availability
- For clinical datasets or third party data, please ensure that the statement adheres to our policy

We provide all data in Supplementary data. The phylogeny we used in the study is shown in Figure 4.

# Field-specific reporting

Please select the one below that is the best fit for your research. If you are not sure, read the appropriate sections before making your selection.

☐ Life sciences  ☐ Behavioural & social sciences  ☒ Ecological, evolutionary & environmental sciences

For a reference copy of the document with all sections, see nature.com/documents/nr-reporting-summary-flat.pdf

# Ecological, evolutionary & environmental sciences study design

All studies must disclose on these points even when the disclosure is negative.

| | |
|---|---|
| Study description | We report an archosauromorph (choristodere) embryo of Ikechosaurus sp. inside a weakly mineralised-shelled egg, from the Lower Cretaceous of northeast China. Phylogenetic comparative analysis on extant and extinct amniotes, including the new fossil, confirm that archosauromorphs displayed EER across oviparity to viviparity, as seen in numerous extant squamates. We show that obligate oviparity evolved multiple times, and viviparity was the primitive reproductive mode, supporting the EER model for origin of the amniotic egg. |
| Research sample | The new specimen (MES-NJU 57004) was collected from yellowish white, thinly laminated tuffaceous mudstone of the Lower Cretaceous Jiufotang Formation (Jehol Biota, ca. 125–120 Ma) in the Lamagou locality adjacent to Chaoyang City, western Liaoning, northeastern China. The specimen is in a recognised, public institution, and available for study by anyone. |
| Sampling strategy | We only have one sample. |
| Data collection | In the phylogenetic analyses we provide all trait data as well as the parameters of the phylogenetic trees and evidence for dating those trees. |
| Timing and spatial scale | Not applied. |
| Data exclusions | No data were excluded. |
| Reproducibility | We provide the tree parameters, specimen dates, and trait data, as well as specification of software used (all of which is standard and available free of charge), so all our analyses can be readily repeated. |
| Randomization | This is applicable to the 'choice of phylogeny' question when running ancestral states analyses, and we provide full details of how thuis was carried out both in the parsimony and Bayesian analyses. |
| Blinding | Not applicable |

Did the study involve field work?   ☐ Yes   ☒ No

# Reporting for specific materials, systems and methods

We require information from authors about some types of materials, experimental systems and methods used in many studies. Here, indicate whether each material, system or method listed is relevant to your study. If you are not sure if a list item applies to your research, read the appropriate section before selecting a response.

## Materials & experimental systems

| n/a | Involved in the study |
|---|---|
| ☒ | Antibodies |
| ☒ | Eukaryotic cell lines |
| ☐ | ☒ Palaeontology and archaeology |
| ☒ | Animals and other organisms |
| ☒ | Human research participants |
| ☒ | Clinical data |
| ☒ | Dual use research of concern |

## Methods

| n/a | Involved in the study |
|---|---|
| ☒ | ChIP-seq |
| ☒ | Flow cytometry |
| ☒ | MRI-based neuroimaging |

## Palaeontology and Archaeology

| | |
|---|---|
| Specimen provenance | The specimen (MES-NJU 57004) was collected from yellowish white, thinly laminated tuffaceous mudstone of the Lower Cretaceous Jiufotang Formation in the Lamagou locality adjacent to Chaoyang City, western Liaoning, northeastern China by the Chinese researchers, according to all aspects of Chinese law |
| Specimen deposition | The specimen is deposited in the School of Earth Sciences and Engineering, Nanjing University to permit free access by other researchers. |

Dating methods | No new dates are provided.

☒ Tick this box to confirm that the raw and calibrated dates are available in the paper or in Supplementary Information.

Ethics oversight | No ethical approval or guidance was required

Note that full information on the approval of the study protocol must also be provided in the manuscript.

