## [Peer Review File · Nature Ecology & Evolution]

Peer Review Information

Journal: Nature Ecology & Evolution

Manuscript Title: Extended embryo retention and viviparity in the first amniotes

Corresponding author name(s): Baoyu Jiang

Editorial Notes:

Reviewer Comments & Decisions:

Decision Letter, initial version:

2nd December 2022

Dear Professor Jiang,

Your manuscript entitled "Extended embryo retention and viviparity in the first amniotes" has now been seen by four reviewers, whose comments are attached. The reviewers have raised a number of concerns which will need to be addressed before we can offer publication in Nature Ecology & Evolution. We will therefore need to see your responses to the criticisms raised and to some editorial concerns, along with a revised manuscript, before we can reach a final decision regarding publication.

Note that we independently contacted reviewer #3, Michel Laurin (he waives his anonymity) to review the manuscript, not being aware that he had reviewed the manuscript previously, but we are happy to use his report and think it provides a useful perspective.

For the remaining comments, while we feel that reviewer 1 provides helpful insight into the role of the new fossil specimen in substantiating your argument, note that reviewers 2 and 4 both question how much it contributes to the paper, emphasising that a lot more clarity is required in the discussion to link both parts of the manuscript into a holistic whole.

We therefore invite you to revise your manuscript taking into account all reviewer and editor comments. Please highlight all changes in the manuscript text file [OPTIONAL: in Microsoft Word format].

* If you have not done so already please begin to revise your manuscript so that it conforms to our

2Article format instructions at <http://www.nature.com/natecolevol/info/final-submission>. Refer also to any guidelines provided in this letter.

[REDACTED]

Nature Ecology & Evolution is committed to improving transparency in authorship. As part of our efforts in this direction, we are now requesting that all authors identified as 'corresponding author' on published papers create and link their Open Researcher and Contributor Identifier (ORCID) with their account on the Manuscript Tracking System (MTS), prior to acceptance. ORCID helps the scientific community achieve unambiguous attribution of all scholarly contributions. You can create and link your ORCID from the home page of the MTS by clicking on 'Modify my Springer Nature account'. For more information please visit www.springernature.com/orcid.

[REDACTED]

Reviewer expertise:

Reviewer #1: choristoderan evolution

Reviewer #2: life history evolution, incl oviparity/viviparity

Reviewer #3: signed report

2Reviewer #4: life history evolution, phylogenetics

Reviewers' comments:

Reviewer #1 (Remarks to the Author):

The paper describes a new fossil specimen from China, interpreted as a late embryo/near-hatchling of the choristoderan reptile *Ikechosaurus*. The identification of the embryo as pertaining to *Ikechosaurus* is convincing. The results presented are novel and are interesting at several levels: they shed light on the varied reproductive strategies of a relatively poorly understood group of extinct reptiles, the choristoderes, and on the skeletal ontogeny of one member of the group – *Ikechosaurus*. The latter observations provide further support for the placement of Choristodera within Archosauromorpha, helping to resolve a longstanding question as to the relationships of this group.

However, the main focus of the paper – taking it beyond specialist appeal - concerns the origins of amniote reproduction, concluding that EER (extended embryo retention) within the mother came before the development of eggs that could be laid on land. Both mammals and squamates show EER, which could be interpreted either as the ancestral condition or independent acquisition in two long-separated amniote lineages. Crocodiles, birds and chelonians, on the other hand, lay eggs at a much earlier ontogenetic stage, with most embryonic development taking place within a nest environment rather than the mother (non-EER). Non-EER could also be the ancestral condition if EER was independently acquired in mammals and squamates.

The authors imply that the recovery of the parchment-shelled egg of a choristodere at an advanced stage of development is suggestive of EER in ancestral archosauromorphs, thereby supporting the view that EER/viviparity was the ancestral amniote condition. However, while I concur with their conclusions with respect to choristodere reproduction, I do not think the authors explain their reasoning clearly enough, or in a logical way.

The conclusion that choristoderes show EER is not based on this new specimen which, as they state (lines 49-51), suggests that *Ikechosaurus* was oviparous and came on land to lay its parchment shelled eggs. This new egg does not demonstrate EER because it could well be an egg that had been developing in a nest for some time prior to fossilisation (as it might in an extant turtle). It is actually previous findings of viviparity in more basal choristoderes, like *Monjurosuchus*, that are indicative of EER in this group. The new specimen of *Ikechosaurus* demonstrates variability in reproductive strategy in choristoderes. Its significance is in supporting the archosauromorph affinities of choristoderes (mainly discussed in SI) and thus, with reference to previous evidence of viviparity in other taxa, that the ancestral condition for choristoderes (and therefore by implication archosauromorphs generally) is EER. The authors do eventually say this (lines 144-148) but should, I think, outline their case more clearly earlier on in the manuscript. As presented, the reader could be left confused as to why the new specimen provides evidence for EER.

My other comments are mostly textural:

1) I think Figure 1 would be more easily interpreted by non-specialists if general clade names (e.g.

3Mammalia, Archosauria etc) were added to some of the small reconstruction figures.

2) Line 44 'absent in archosauromorphs, including turtles etc' – should read archelosaurs. Chelonians are not archosauromorphs.

3) In figure 2 – write explanation of EDS out in full in the caption so the reader doesn't have to search for it in the methods. There is space to do this in the caption.

4) On line 86 of the main text – flattened rather than flatted

5) In the caption to Figure 3 – exopterygoid should be ectopterygoid

6) On line 135 (main text) ...'egg of viviparous species IS lacking ...

7) On line 463 (main text) ... 'of the' ... is duplicated

8) SI – p.2 paragraph 1: the bone between the maxilla and prefrontal is either spelt lacrimal or lachrymal. In this paragraph it is first spelt lacrymal (incorrectly), then lacrimal on the next line, then lacrymal on the line below. It is spelt lacrimal in the main text (caption to Fig.3). Please correct this in the description.

9) SI – p.2, para 2, line 3: ...'with a single nasal in between articulating and the premaxilla' – does not make sense. English-speaking co-author to correct.

10) SI p.3, para 2, line 1 and 3, add 'the' in front of left forelimb (line 1) and humerus (line 3)

11) SI, p3, para 2, line 8 – the sentence starting 'The femur..' is awkwardly phrased. At the very least, add a comma after 'ossification'.

12) SI, p.3, para 3 –'embryogenetic'? A more common term is ontogenetic

13) SI, p.4, para 1, line 9, '.....in which it remains' – define 'it'. In this context, 'it' presumably refers back to 'sutures' – in which case 'it' should be 'they'. Better, however, to write 'sutures' here, given that sutures were last mentioned several lines above.

14) Ditto – on the next line to that mentioned in (11) – 'it is often closed' should be 'they are often closed'

Reviewer #2 (Remarks to the Author):

Review on NATECOLEVOL-221117892

Extended embryo retention and viviparity in the first amniotes
by Jiang et al.

The amniote egg is a key innovation for conquering the terrestrial realm. In this context it is an important question where and how the foetal membranes of the amniotic egg evolved, i.e., whether the ancestor of amniotes laid eggs on land that were similar to that of amphibians with direct development and in the course of evolution foetal membranes were gradually acquired (terrestrial model) or that membranes appeared in the oviduct of the ancestor and were linked to egg retention (EER).

The paper of Jiang et al. presents a new find of an embryo inside a parchment-shelled egg of the choristodere *Ikechocaurus spec.*. The authors included this new find in an ancestral state reconstruction analysis (ASR) in order the find support for the terrestrial or the EER model on the ancestral state. Their ACR analysis is very thoroughly with respect to errors in the phylogenetic tree used, with respect to the method used for its calculation, position of taxonomic groups in the tree and

4divergence time. The authors also take care of uncertainties in trait values of the species *Mesosaurus tenuidens* in which viviparity and oviparity co-occur.

As far as I have understood their methods correctly, they used two approaches on modelling the reproductive mode of taxa, egg shell mineralisation and EER. The methodological challenging task of their study is that these characters do not evolve independently. To cope with that the authors used hidden states in a single structured markov model in combination with both a maximum-likelihood and Bayesian algorithm. Their second approach was parsimony-based ASR. The evolutionary models tested by both approaches were different and only the maximum-likelihood algorithm and the Bayesian algorithm revealed information on goodness-of-fit of data to evolutionary models tested. The major finding of the ASR analysis is that the all-rates-different model always reveals the best evolutionary model and that the ancestor of amniotes showed EER and was viviparous.

Contrary to the presentation of the ASR analysis, the text of the manuscript on introducing the question and presenting *Ikechosaurus* spec. is well-written.

As I'm no specialist on Choristodera, I have only a number of major comments on the study of Jiang et al. that refer to the ASR analysis.

(1) I do not see, why the exceptional find of *Ikechosaurus* is mandatory for the ASR analysis.

An inspection of the tree and the character states of the terminal species shows that the tree is in principle dichotomous with only the Archelosauria (of which *Ikechosaurus* is member of) not clearly showing the reconstructed ancestral viviparity and EER. The best evolutionary model is the all-rates-different model, which can easily render the different evolutionary courses in the two parts of the tree and converge to the ancestral state as nearly all basal taxa show this mode.

I would be curious to see the results of a similar ACR analysis in which *Ikechosaurus* is omitted. I guess the authors will arrive at the identical results.

Moreover, *Ikechosaurus* is only used once in the discussion (line 267-271). This underlines that *Ikechosaurus* is not mandatory for the manuscript.

(2) Although I understand that the number of species that are suitable for the ASR analysis is small, I think the set of species used in the analysis is biased. For example, all extinct marine reptiles used are viviparous, although oviparous species existed. However, this old group is very influential to the reconstructed ancestral state and pushes the characters towards viviparity with EER.

Would an omission of marine reptiles from the dataset reveal the identical result? Would an ASR analysis only conducted with extant species reveal the identical result? Overall, I would like to see some analysis tackling the influence of the dataset itself on the reconstructed ancestral state of amniotes.

(3) The material and method section is not well structured with respect to the ASR analysis. I'm really not sure whether I correctly understood what the authors did. After referring to the supplemental material it became somewhat clearer to me. I suggest to completely reorganize this text regarding the evolutionary models analysed, the algorithms used and the phylogenetic trees considered. The heavy usage of acronyms that are not all defined or the definition is found later in the text increased my confusion. Please keep in mind that not all readers of *NatEcolEvol* are specialists in ASR analysis.

I think Tables and Figures in the Supplement on what you did would be very helpful (e.g., Fig. 3). The

5complexity of all your analysis is very high. A further option would be to remove the results from the parsimony-based ASR which has no information on goodness-of-fit of models. Moreover, the three parsimony models do less well capture character transitions (see Lines 515-519). The tables with results on ASR analysis are huge and as models are shown on different pages they are difficult to compare. Would it be an option to use a larger piece of paper so that everything is on a single page? I'm pretty sure printing is not an option for this large document.

Specific points

Line 10: extant amniotes, turtles, crocodylans, and birds. Turtles, crocodylans, and birds are amniotes.

Line 12: only a few extant mammals show oviparity

Line 87: computer

Line 177: What are Sankoff matrices? Sankoff parsimony is an algorithm? Please provide more information. What's about the large number of trees that you used in ASR analysis? I think you should at least prepare the reader to understand the different analysis in the Supplementary Material (point 2) and follow your discussion. Replace framework by frameworks.

Line 183: Figure 4. Please increase the font size in the time bar and provide the number for extinct and extant species. The two blue colors (present and membrane-shelled eggs) are difficult to distinguish. Without more information on your methods (Line 177) FBD and CARD_sw are difficult to understand. What is Diapsida 1 and Diapsida 2?

Line 197: both characters. Which?

Line 198-203: What are viv.-ML, Diapsida s.l.? Is there a difference between Diapsida s.l. and Diapsida 1(2)?, refer percentages to numbers of trees tested?

Line 207, 215: What's MP (BT)-based ancestral state reconstruction?

Line 237-239: As far as I understand your ASR methods correctly these start from tips and trace states back to the root. Based on that, I find it problematic that later diverging clades are incorrectly modelled. Shouldn't the inaccuracy increase from the tips to the root? Given how the ancestral state is derived, why should I believe that an evolutionary model that is not correct for the extant species is correct for extinct ones?

Line 255: not all saupterygians are viviparous. Please discuss this.

Line 264: isolated eggs are difficult to assign to species. This is also an important taphonomic bias.

Line 450: What's the absolute time of the anchor species Mesosaurus tenuidens?

Line 451: why $\gamma = 3$?

Line 458: inspired by ref. 52, inspired in terms of what?

Line 477 and other: A table in the Supplement on all these models, their acronyms and how they work would be comfortable for a reader.

Line 503-504: please provide the name of the function. Phytools provides a lot of beautiful graphs. I guess you plotted ACE values as colored circles. What are final plots?

Line 515-523: Please explain this in more detail. What are the evolutionary models tested, why these models?

Line 525: BatesTraits is this a software, an algorithm?

Line 549: You considered more alternative phylogenies besides of dropping Mesosaurus tenuidens. What's about a table which phylogenies were considered.

Supplement

Table S1: please provide the information extinct/extant

Reviewer #3 (Remarks to the Author):

This draft is nearly ready for publication. I had reviewed at least two previous versions, for Nature. I found two sentences that need to be toned down to avoid sounding like you over-interpret your findings. Readers risk becoming suspicious if they get the impression that you want to sell moderate evidence as strong evidence. One sentence needs corrections and probably should be split, and another one contains a wrong taxonomic statement (see below). All this should take only minutes to fix.

Lines 20-21: "We show that non-EER oviparity evolved multiple times, and viviparity was the primitive reproductive..." To "show" is a strong statement in science. I would start this sentence by "Our analyses suggest that", or "Our analyses strongly favor..."

Line 41: "Strong evidence for the EER model is the near absence of fossils of...". No, negative evidence plays a large role here (no evidence of the reproductive mode of any of the dozens of species of Paleozoic terrestrial amniotes), so at least remove the "strong". I would put "circumstantial" there, or "weak".

Line 47: "As EER occurs widely among extant lizards and snakes (lepidosauromorphs)". Actually, "lizards" and snakes form Squamata, not Lepidosauromorpha, which is a much larger clade (with Sphenodon and many extinct taxa). Also, in such a paper, I would avoid paraphyletic taxa; why not just state "squamates"? I don't think that most potential readers of this paper will be puzzled by this word.

7Lines 134-138: "It could be argued that the described choristodere specimen is an incomplete viviparous egg^{27,28}, but in extant squamates, the egg of viviparous species lacking the calcified layer, but is enveloped with a very thin organic layer (commonly less than 10 μm), whereas oviparous squamates lay eggs with a calcified outer layer and a relatively thick organic layer (usually over 30 μm)^{29,30}, as here. » This sentence is too long, has two « but », and has a grammatical mistake "lack" is needed, rather than "lacking".

Best wishes,

Michel Laurin

Reviewer #4 (Remarks to the Author):

This paper reports two main results: the description of a fossil embryo/fetus with eggshell, and a phylogenetic analysis of parity mode. The fossil is identified as a choristodere embryo of *Ikechosauros*. I can not attest to the quality of this part of the paper since I am not a paleontologist, but given the expertise of the authors, I am willing accept their conclusions on the identity of the embryo. The phylogenetic analysis strongly suggests that extended embryo retention (EER) and may be even viviparity was the ancestral amniote reproductive mode. This conclusion is in contrast to the standard model which claims that the characteristic traits of the amniote egg are adaptations to terrestrial oviparity. The EER model is a very interesting novel development in our understanding of vertebrate reproductive biology and has been a minority position for most of the 20th century. If confirmed a model where the amniotic extraembryonic membranes evolved during egg retention would completely change the story about the functional significance of these structures. In a way, such a model would be rather attractive, because dropping an an-amniotic egg on land and hoping for the amnion to evolve to protect the embryo, does not seem to be a winning strategy. Teaching this model over the years I always felt unhappy, given that anyone who ever had alligator eggs in his/her lab knows that alligator eggs are NOT desiccation resistant, to the contrary!

But if amniote egg structures evolved for fetal-maternal communication, this would make a lot of sense. It would also explain why even avian extra-embryonic tissues are producing steroid hormones and other signals, which could be residues of the time when they had to communicate with the mother during extended embryo retention. This is a potentially important paper, but see major comment 1 below.

Major comments:

1) While the phylogenetic analysis is very well done and conservative in its assumptions, it is curious that on the synapsid side of the amniote phylogeny there are no non-mammalian fossils represented, as far as I can see. This could be due to a lack of developmental stages in the pre-mammalian synapsid fossil record, and I do not know whether that is the case. At the very least I would encourage the authors to state what is and is not known from pre-mammalian synapsid fossils. This is an important issue, because the current taxon sampling is heavily biased in favor of

8Reptilia/Sauropsida, which could give the impression of strong support of what is a reptilian rather than an amniote ancestral characteristic.

2) The description of the novel embryo is only loosely related to the main result of the paper, which is based on the phylogenetic analysis of previously published data. The new fossil is way too young to be of significance for the phylogenetic analysis. Of course it is nice to have statistical inferences and real data combined in a paper, but I just want to point out that the connection between the two parts of the paper is not very strong.

Minor comments:

This paper is very well written and cites older and even obscure but relevant literature, and I applaud the authors for that.

Günter Wagner, Yale University

*****END*****

Author Rebuttal to Initial comments

Reviewers' comments:

Reviewer #1 (Remarks to the Author):

The paper describes a new fossil specimen from China, interpreted as a late embryo/near-hatchling of the choristoderan reptile *Ikechosaurus*. The identification of the embryo as pertaining to *Ikechosaurus* is convincing. The results presented are novel and are interesting at several levels: they shed light on the varied reproductive strategies of a relatively poorly understood group of extinct reptiles, the choristoderes, and on the skeletal ontogeny of one member of the group – *Ikechosaurus*. The latter observations provide further support for the placement of Choristodera within Archosauromorpha, helping to resolve a longstanding question as to the relationships of this group.

Many thanks!

9However, the main focus of the paper – taking it beyond specialist appeal - concerns the origins of amniote reproduction, concluding that EER (extended embryo retention) within the mother came before the development of eggs that could be laid on land. Both mammals and squamates show EER, which could be interpreted either as the ancestral condition or independent acquisition in two long-separated amniote lineages. Crocodiles, birds and chelonians, on the other hand, lay eggs at a much earlier ontogenetic stage, with most embryonic development taking place within a nest environment rather than the mother (non-EER). Non-EER could also be the ancestral condition if EER was independently acquired in mammals and squamates. The authors imply that the recovery of the parchment-shelled egg of a choristodere at an advanced stage of development is suggestive of EER in ancestral archosauromorphs, thereby supporting the view that EER/viviparity was the ancestral amniote condition. However, while I concur with their conclusions with respect to choristodere reproduction, I do not think the authors explain their reasoning clearly enough, or in a logical way. The conclusion that choristoderes show EER is not based on this new specimen which, as they state (lines 49-51), suggests that *Ikechosaurus* was oviparous and came on land to lay its parchment shelled eggs. This new egg does not demonstrate EER because it could well be an egg that had been developing in a nest for some time prior to fossilisation (as it might in an extant turtle). It is actually previous findings of viviparity in more basal choristoderes, like *Monjurosuchus*, that are indicative of EER in this group. The new specimen of *Ikechosaurus* demonstrates variability in reproductive strategy in choristoderes. Its significance is in supporting the archosauromorph affinities of choristoderes (mainly discussed in SI) and thus, with reference to previous evidence of viviparity in other taxa, that the ancestral condition for choristoderes (and therefore by implication archosauromorphs generally) is EER. The authors do eventually say this (lines 144-148) but should, I think, outline their case more clearly earlier on in the manuscript. As presented, the reader could be left confused as to why the new specimen provides evidence for EER.

Thanks for these constructive suggestions. We clarify our new finding in the introductory portion.

My other comments are mostly textual:

- 1) I think Figure 1 would be more easily interpreted by non-specialists if general clade names (e.g. Mammalia, Archosauria etc) were added to some of the small reconstruction figures.
Added.
- 2) Line 44 ‘absent in archosauromorphs, including turtles etc’ – should read archelosaurs. Chelonians are not archosauromorphs.
Corrected.
- 3) In figure 2 – write explanation of EDS out in full in the caption so the reader doesn’t have to search for it in the methods. There is space to do this in the caption.

Revised.

4) On line 86 of the main text – flattened rather than flatted

Corrected.

5) In the caption to Figure 3 – exopterygoid should be ectopterygoid

Corrected.

6) On line 135 (main text)'egg of viviparous species IS lacking ...

Corrected.

7) On line 463 (main text) ... 'of the' ... is duplicated

Corrected.

8) SI – p.2 paragraph 1: the bone between the maxilla and prefrontal is either spelt lacrimal or lachrymal. In this paragraph it is first spelt lacrymal (incorrectly), then lacrimal on the next line, then lacrymal on the line below. It is spelt lacrimal in the main text (caption to Fig.3). Please correct this in the description.

Corrected.

9) SI – p.2, para 2, line 3: ...'with a single nasal in between articulating and the premaxilla' – does not make sense. English-speaking co-author to correct.

Corrected.

10) SI p.3, para 2, line 1 and 3, add 'the' in front of left forelimb (line 1) and humerus (line 3)

Corrected.

11) SI, p3, para 2, line 8 – the sentence starting 'The femur..' is awkwardly phrased. At the very least, add a comma after 'ossification'.

We reorganized this sentence.

12) SI, p.3, para 3 –'embryogenetic'? A more common term is ontogenetic.

Corrected.

13) SI, p.4, para 1, line 9, '.....in which it remains' – define 'it'. In this context, 'it' presumably refers back to 'sutures' – in which case 'it' should be 'they'. Better, however, to write 'sutures' here, given that sutures were last mentioned several lines above.

Corrected.

14) Ditto – on the next line to that mentioned in (11) – 'it is often closed' should be 'they are often closed'

Corrected.

Reviewer #2 (Remarks to the Author):

Review on NATECOLEVOL-221117892

Extended embryo retention and viviparity in the first amniotes

by Jiang et al.

The amniote egg is a key innovation for conquering the terrestrial realm. In this context it is an important question where and how the foetal membranes of the amniotic egg evolved, i.e., whether the ancestor of amniotes laid eggs on land that were similar to that of amphibians with direct development and in the course of evolution foetal membranes were gradually acquired (terrestrial model) or that membranes appeared in the oviduct of the ancestor and were linked to egg retention (EER). The paper of Jiang et al. presents a new find of an embryo inside a parchment-shelled egg of the choristodere *Ikechosauros spec.*. The authors included this new find in an ancestral state reconstruction analysis (ASR) in order the find support for the terrestrial or the EER model on the ancestral state. Their ACR analysis is very thoroughly with respect to errors in the phylogenetic tree used, with respect to the method used for its calculation, position of taxonomic groups in the tree and divergence time. The authors also take care of uncertainties in trait values of the species *Mesosaurus tenuidens* in which viviparity and oviparity co-occur. As far as I have understood their methods correctly, they used two approaches on modelling the reproductive mode of taxa, egg shell mineralisation and EER. The methodological challenging task of their study is that these characters do not evolve independently. To cope with that the authors used hidden states in a single structured markov model in combination with both a maximum-likelihood and Bayesian algorithm. Their second approach was parsimony-based ASR.

12The evolutionary models tested by both approaches were different and only the maximum-likelihood algorithm and the Bayesian algorithm revealed information on goodness-of-fit of data to evolutionary models tested. The major finding of the ASR analysis is that the all-rates-different model always reveals the best evolutionary model and that the ancestor of amniotes showed EER and was viviparous. Contrary to the presentation of the ASR analysis, the text of the manuscript on introducing the question and presenting *Ikechosaurus spec.* is well-written.

Thank you for the positive comments; we attend below to the lack of clarity in Methods descriptions.

As I'm no specialist on Choristodera, I have only a number of major comments on the study of Jiang et al. that refer to the ASR analysis.

(1) I do not see, why the exceptional find of *Ikechosaurus* is mandatory for the ASR analysis.

An inspection of the tree and the character states of the terminal species shows that the tree is in principle dichotomous with only the Archelosauria (of which *Ikechosaurus* is member of) not clearly showing the reconstructed ancestral viviparity and EER. The best evolutionary model is the all-rates-different model, which can easily render the different evolutionary courses in the two parts of the tree and converge to the ancestral state as nearly all basal taxa show this mode. I would be curious to see the results of a similar ACR analysis in which *Ikechosaurus* is omitted. I guess the authors will arrive at the identical results. Moreover, *Ikechosaurus* is only used once in the discussion (line 267-271). This underlines that *Ikechosaurus* is not mandatory for the manuscript.

We understand the concern, but we argue that the fossil is important for several reasons, especially that it shows what the palaeontological data on reproductive mode look like with a concrete example, but one that confirms multiple modes in a single genus. This is known among some living squamates, where species within a genus can be either viviparous or oviparous, but here's an unexpected fossil example. The third reason is that most textbook assumptions are that Archosauromorpha are entirely oviparous, so this and a few other recent examples show diversity of reproductive modes in this major clade, and it was these observations that set of the train of thought with us, and others, that perhaps the mineralized egg was not the first reproductive mode to appear in Amniota, nor in any of its main constituent clades.

(2) Although I understand that the number of species that are suitable for the ASR analysis is small, I think the set of species used in the analysis is biased. For example, all extinct marine reptiles used are viviparous, although oviparous species existed. However, this old group is very influential to the reconstructed ancestral state and pushes the characters towards viviparity with EER. Would an omission of marine reptiles from the dataset reveal the identical result? Would an ASR analysis only conducted with extant species reveal the identical result? Overall, I would like to see some analysis tackling the influence of the dataset itself on the reconstructed ancestral state of amniotes.

We thank the reviewer for the suggestions. We acknowledge that the 'macroevolutionary study' section in the main portion of the manuscript and subsequent method section, were not particularly well structured, for which we apologize. Consequently, we would like to take the opportunity to clarify the analyses that we have already undertaken. We have conducted over 100,000 separate analyses using various perturbations for our ancestral state reconstruction. The ancestral state reconstruction (note, that several of these approaches are multiplicative, not additive, e.g., 8 max. likelihood models are used for all 4 time-scaling methods, etc.) was run:

- a) for 4 different time-scaling approaches, each of them resulting in 100 time-scaled trees
- b) 4 completely different topologies
- c) with and without taxa (*Mesosaurus*) deemed to have an extraordinary influence on the ancestral state estimates
- d) allowing for uncertainty in input tip states
- e) using three different ancestral state estimation approaches (maximum likelihood, parsimony, Bayesian)
- f) 8 + 2 likelihood models, 3 + 1 parsimony models, 16 (+16 for constrained models) + 4 Bayesian models
- g) allowing for evolutionary rate heterogeneity in Bayesian models
- h) with and without constraining node states of Lepidosauria and Squamata

Irrespective of these perturbations, our ancestral state reconstructions remain qualitatively the same.

Regarding the influence of fossils in the dataset: By running the analyses with and without *Mesosaurus* we tested for the potential extraordinary influence of the taxon on the ancestral state reconstruction. In a similar vein of thought we changed the position of extinct marine reptiles across different topologies. We tested three alternative phylogenetic positions: sister taxa to Archelosauria, sister to Archosauromorpha, and sister to Lepidosauria. Importantly, all alternative tree topologies increased the distance of marine reptiles relative to the root of the tree, thus decreasing the influence of the taxa with respect to the root ancestral state reconstruction.

To specifically address the reviewers' points, we ran an additional set of analyses where we excluded the extinct marine reptiles from our dataset. The results for maximum likelihood (Tables S11–S14), Bayesian (Tables S36–S39), and parsimony (Table S23) approaches can be found in the supplement. The EER ancestral state reconstructions remain unaffected by the removal of extinct marine reptiles. As for the reconstruction of reproduction mode and eggshell mineralisation: We find that this change has little effect on ancestral state estimates of the best fitting models for both maximum likelihood and Bayesian approaches. Qualitatively the results of the best fitting models are the same as the ones which include extinct marine reptiles. We still recover viviparity at the origin of Amniota. The removal of extinct marine reptiles, however, does affect the ancestral state reconstructions of models that do not belong to the best-fitting category (AIC difference > 2 ; log BF difference > 2), showcasing increased uncertainty in the ancestral state estimates between different models. Furthermore, some of the models for which the states of both Lepidosauria and Squamata were fixed a priori (Tables S36–S39) also show more uncertainty in the ancestral state reconstructions if the extinct marine reptiles are removed.

We add the following lines to the manuscript (lines 271-275):

“Completely removing the extinct marine reptiles from the analyses has little impact on the results, with only models with a worse fit and some of the models for which the states of both Lepidosauria and Squamata were fixed a priori showing more uncertainty in the ancestral state reconstructions (Tables S11–S14; S36–S39).”

And furthermore we revise the methods as follows (lines 572-576):

“To test the robustness of our results we reran all our analyses dropping *Mesosaurus tenuidens* and extinct marine reptiles, respectively, from our input phylogeny.”

(3) The material and method section is not well structured with respect to the ASR analysis. I'm really not sure whether I correctly understood what the authors did. After referring to the supplemental material it became somewhat clearer to me. I suggest to completely reorganize this text regarding the evolutionary models analysed, the algorithms used and the phylogenetic trees considered. The heavy usage of acronyms that are not all defined or the definition is found later in the text increased my confusion. Please keep in mind that not all readers of NatEcolEvol are specialists in ASR analysis.

We have included a short overview of the ancestral states estimation methods in the main text, which neatly summarizes the more comprehensive methods section:

“We conducted an exhaustive multiplicative set of ancestral states analyses, accounting for: i) different phylogenetic time-scaling methods; ii) alternative tree topologies; iii) exclusion of key fossils; iv) different models and optimization criteria; v) among lineage rate heterogeneity; vi) constraining extant node states based on previous studies. Together, these totaled over 100,000 individual analyses.”

We have also added subheadings to the ‘Phylogenetic Macroevolutionary Analysis’ section of the methods, which should aid readers in understanding this rather long section. Finally, we provide a new version of figure S3, which aims at visualizing the different models employed in our analysis. We hope this makes the analyses clearer.

I think Tables and Figures in the Supplement on what you did would be very helpful (e.g., Fig. 3). The complexity of all your analysis is very high. A further option would be to remove the results from the parsimony-based ASR which has no information on goodness-of-fit of models. Moreover, the three parsimony models do less well capture character transitions (see Lines 515-519). The tables with results on ASR analysis are huge and as models are shown on different pages they are difficult to compare. Would it be an option to use a larger piece of paper so that everything is on a single page? I'm pretty sure printing is not an option for this large document.

We cannot remove the parsimony results, since another reviewer insisted on running and expanding the parsimony section. Our goal is not to discuss different methods of ancestral state reconstruction. Parsimony offers the possibility to run a ‘branch-length-free approach’, which, however (as pointed out by the reviewer), suffers from it non being model-based – therefore model comparisons as with maximum likelihood or Bayesian approaches are not possible.

Regarding the tables – unfortunately we are constrained by the PDF constraints for supplements given by Nature Ecology & Evolution. We tried to circumvent the issue, by keeping the models that usually would be found among the best-fitting ones on the same page. Furthermore we also colored the best-fitting models in green (absolute best model for a specific perturbation of the dataset) and gray (best-fitting models that cannot be distinguished from the absolute best model based on AIC or log BF value).

Specific points

Line 10: extant amniotes, turtles, crocodilans, and birds. Turtles, crocodilans, and birds are amniotes.

Corrected.

Line 12: only a few extant mammals show oviparity

Agreed, and explained later; we have to keep the Abstract brief, and avoid subclauses and explanations that are given in the main paper.

Line 87: computer

Not sure what is intended here.

Line 177: What are Sankoff matrices? Sankoff parsimony is an algorithm? Please provide more information. What's about the large number of trees that you used in ASR analysis? I think you should at least prepare the reader to understand the different analysis in the Supplementary Material (point 2) and follow your discussion. Replace framework by frameworks.

Sankoff matrices are cost matrices that allow us to put a penalty on character state transitions. Our maximum likelihood and Bayesian structured Markov models (SMM) forbid certain character transition (e.g., to model a switch-on dependency; see also supplementary fig. S3). Sankoff matrices allow us to implement an approach for the amalgamated character that is akin to the maximum likelihood and Bayesian approaches but within parsimony. A transition that requires a single (parsimony) step (step cost = 1) represents a transition that is allowed in our model (a step cost of 0 represents a transition from one state to itself). Setting the cost for transition of these states higher makes it less likely that the respective transition will be recovered as the most parsimonious one. In fact, if we set the cost of transition high enough (for our dataset and in our case we used a value of 100), such a transition will practically never be recovered. By increasing the cost of transition to such a high value we are therefore able to forbid certain transitions from occurring in a parsimony setting, very similar in this regard to the maximum likelihood and Bayesian SMMs.

A detailed description of Sankoff matrices is found in our reference 56. Ref. 54 gives a less technical explanation of Sankoff matrices and their merits. We realize, however, that these references were not found adjacent to the term Sankoff, therefore making it difficult to spot the reference. We revised the text accordingly.

We revised the manuscript as follows:

Lines 181-182 (removing 6 and keeping framework):

“We also applied an analogous approach within a parsimony framework using Sankoff (cost) matrices.”

Lines 538-540 (adding in text references and further explanation of the step costs required to prevent certain character state transitions within parsimony):

“[...] using Sankoff (cost) matrices^{54, 56}. The number of required steps for transitions that were forbidden in the model was set to a value (step cost = 100) that would make it practically impossible for the transition to occur.”

Ref. 54 and 56:

54. Maddison, W. P. Missing data versus missing characters in phylogenetic analysis. *Syst. Biol.* 42, 576–581 (1993).

56. Sankoff, D. & Rousseau, P. Locating the vertices of a steiner tree in an arbitrary metric space. *Math. Program.* 9, 240–246 (1975).

Line 183: Figure 4. Please increase the font size in the time bar and provide the number for extinct and extant species. The two blue colors (present and membrane-shelled eggs) are difficult to distinguish. Without more information on your methods (Line 177) FBD and CARD_sw are difficult to understand. What is Diapsida 1 and Diapsida 2?

Diapsida s.l. (= Diapsida 1) represent Diapsida sensu lato, i. e. including extinct marine reptiles. Diapsida s.s. represent Diapsida sensu stricto (= Diapsida 2), i. e. Diapsida as defined by extant clades only (excluding extinct marine reptiles). We changed figure 4 accordingly.

The revised caption for Supplementary fig. S3 provides additional information on the CARD model. The acronym for the fossilised birth-death tip-dating method (FBD) is introduced in line 453.

Line 197: both characters. Which?

Manuscript revised to (lines 204-205):

“The result is unequivocal for both the amalgamated character (reproduction mode + eggshell mineralisation) and EER presence.”

Line 198-203: What are viv.-ML, Diapsida s.l.? Is there a difference between Diapsida s.l. and Diapsida 1(2)?, refer percentages to numbers of trees tested?

viv.-ML is the acronym for viviparity-maximum likelihood. We define the acronym ML in the methods section but to improve clarity, we change the first appearance of the acronym in the manuscript and write it in full (Lines 205-213 revised to):

Viviparity with EER dominates the deeper nodes, being the most likely condition for the roots of Amniota (mean marginal maximum likelihood (ML)-based ancestral state across 100 trees for best-fitting model and other models whose AIC difference is <2 compared to best-fitting model for viviparity-ML: 99.5–100%; EER-ML: 99.8–100%), Reptilia (viv.-ML: 98.5–100%; EER-ML: 100%), Diapsida s.l. (viv.-ML: 100%; EER-ML: 100%), Archelosauria (viv.-ML: 98.5–100%; EER-ML: 99.6–100%), and Archosauromorpha (viv.-ML: 99.4–100%; EER-ML: 99.8–100%), irrespective of which time-scaling approach or best-fitting model (including models, whose AIC differs from the best-fitting model by less than 2) is considered (Tables S3–S6).

Diapsida s.l. (= Diapsida 1) represent Diapsida sensu lato, i. e. including extinct marine reptiles. Diapsida s.s. represent Diapsida sensu stricto (= Diapsida 2), i. e. Diapsida as defined by extant clades only (excluding extinct marine reptiles). We changed figure 4 accordingly.

Percentages refer to the range in ancestral state reconstructions of the best-fitting models as reported in the Supplementary Tables (in the example above that would be Tables S3-S6).

Line 207, 215: What's MP (BT)-based ancestral state reconstruction?

We define the acronyms MP and BT in the methods section, but for clarity we now write them in full as well for the first appearance in the manuscript:

Lines 214-219:

“Using a different ancestral state reconstruction method also does not change the main conclusions: ten out of eighteen maximum parsimony (MP)-based ancestral state reconstructions recover viviparity at the origin of Amniota, with the remaining reconstructions mostly favouring membrane-shelled eggs as the ancestral state of amniotes (all MP-based reconstructions using ACCTRAN favour viviparity at the origin of Amniota; Tables S21–S26).”

Lines 220-221:

“Bayesian results (BT) are consistent with ML-based ancestral state reconstructions (Tables S28–S31a-d; Supplementary Fig. 4).”

Line 237-239: As far as I understand your ASR methods correctly these start from tips and trace states back to the root. Based on that, I find it problematic that later diverging clades are incorrectly modelled. Shouldn't the inaccuracy increase from the tips to the root? Given how the ancestral state is derived, why should I believe that an evolutionary model that is not correct for the extant species is correct for extinct ones?

The reviewer is correct in that we should expect uncertainty in ancestral state analyses to propagate towards the root. However, there are two key factors that are worth keeping in mind. Firstly, tree shape dictates how uncertainty propagates in both parsimony and likelihood-based ancestral state analyses. Asymmetrical trees tend to have more certainty in their ancestral state estimates because terminal taxa can more directly inform internal node states. Secondly, under likelihood ancestral state estimation, taxa that sit upon short branches will provide more information with respect to their parent node than those that sit upon long branches. This is true of fossil taxa in time-scaled trees. The supertree(s) we use in our ancestral states analyses are both asymmetrical and include lots of fossils that sit upon short branches. As such, our tree is (by design) predisposed to provide maximum information towards the root.

We do not state that the ancestral state estimates for later diverging clades are modelled incorrectly. Instead, we only state (based on our data), that the results for younger clades are less consistent when comparing different models than the results for earlier diverging clades closer to the root. This in itself is not meant to indicate that the respective results are necessarily wrong. It is only an observation that the

20range of ancestral state estimates is larger for these younger clades. There are two probable explanations for this:

(1) Character states are more variable in later diverging clades than in the nodes and tips close to the root (a good example is, indeed, Lepidosauria, which has also attracted attention before in the literature (see, e.g., ref. 4-6).

(2) The focus of this analysis and dataset is not on the fine details of reproduction and EER distribution within Squamata, Lepidosauria, or chelonians (or other younger amniote clades). Including additional extant taxa could help in elucidating the ancestral states within these clades, but it would have little impact on the deeper nodes (see also Li et al. 2008 for further information). Instead, our focus lies on the ancestral states at the origin of Amniota and of deeper nodes within Amniota. Our sampling strategy reflects our motivation to understand the ancestral states of early diverging nodes within Amniota. As such, we have sampled all fossil taxa for which reproduction mode can be reasonably inferred. We have also sampled representatives of living groups spanning the phylogenetic diversity of Amniota. Adding more extant taxa could therefore potentially increase consistency in the ancestral state estimates of younger clades (but see the controversy surrounding extant squamates mentioned before) – but it would have little impact on our results and is completely outside the scope of this analysis.

4. Pyron, R. A. & Burbrink, F. T. Early origin of viviparity and multiple reversions to oviparity in squamate reptiles. *Ecol. Lett.* 17, 13–21 (2014).
5. King, B. and Lee, M. S. Y. Ancestral state reconstruction, rate heterogeneity, and the evolution of reptile viviparity. *Syst. Biol.* 64, 532–544 (2015).
6. Wright, A. M., Lyons, K. M., Brandley, M. C. & Hillis, D. M. Which came first: the lizard or the egg? Robustness in phylogenetic reconstruction of ancestral states. *J. Exp. Zool. (Mol. Dev. Evol.)* 324B, 504–516 (2015).

Li, Guoliang, Mike Steel, and Louxin Zhang. "More taxa are not necessarily better for the reconstruction of ancestral character states." *Systematic Biology* 57, no. 4 (2008): 647-653.

Line 255: not all saupterygians are viviparous. Please discuss this.

We revised this to “some sauropterygians”.

Line 264: isolated eggs are difficult to assign to species. This is also an important taphonomic bias.

Yes, we added this.

Line 450: What's the absolute time of the anchor species *Mesosaurus tenuidens*?

Yes, we added this: Early Permian, Kungurian, 278.4 Ma

Line 451: why gamma = 3?

Gamma = 3 produces a distribution that best matches our prior belief about the quality of the amniote fossil record. Specifically, such a distribution suggests that age of the oldest fossils member of a clade is likely to be close to the true age of that clade, but is unlikely to equal to the true age of that clade. Below is an example of the calibration for crown-group mammals used in our analysis. The red and blue lines indicate the hard minimum and soft maximum ages from Benton et al 2015 (ref 51).

Hard minimum = 164.9, soft maximum = 201.5, shape = 3, percentile = 0

Line 458: inspired by ref. 52, inspired in terms of what?

Ref. 52 is used by the paleotree package and specifically the `createMrBayesTipDatingNexus()` function as an example for recommended best practices in tip-dating analyses. FBD and clock priors used by the function are derived from this work.

We revised the manuscript to (lines 474-475):

“The default FBD and clock priors (see ref. 52) provided by ‘`createMrBayesTipDatingNexus`’ were kept.”

2252. Matzke, N. J. & Wright, A. Inferring node dates from tip dates in fossil Canidae: the importance of tree priors. *Biol. Lett.* 12, 20160328 (2016).

Line 477 and other: A table in the Supplement on all these models, their acronyms and how they work would be comfortable for a reader.

We provide an updated supplementary fig. S3 with an expanded caption, which also explains the respective acronyms.

Line 503-504: please provide the name of the function. Phytools provides a lot of beautiful graphs. I guess you plotted ACE values as colored circles. What are final plots?

We revised the manuscript to (lines 521-524):

“We then calculated the mean marginal ancestral states of the best fitting model for each set of 100 input trees, which were plotted on a consensus tree generated using the ‘consensus.edges’ function of the R package ‘phytools’⁵⁸. Plots were generated using the R package ‘strap’⁵⁹.”

Line 515-523: Please explain this in more detail. What are the evolutionary models tested, why these models?

We now provide an updated supplementary fig. S3 with expanded captions which explains the different evolutionary models employed in a maximum likelihood and Bayesian setting. For parsimony it is not possible to estimate different transition rate models (since parsimony is a branch-length free approach), but it is possible to use models that assume character independence (ind) or switch-on dependency (sw). We have changed lines 534-544 as detailed for the question regarding Sankoff matrices, further expanding on Sankoff matrices and adding also new inline references to the relevant literature. The reasoning for the choice of models is the same as for the maximum likelihood and Bayesian ones and is detailed in lines 485-501.

Line 525: BatesTraits is this a software, an algorithm?

23Revised manuscript (lines 545-546):

Furthermore, we also ran a Bayesian (BT) ancestral state reconstruction using the software package BayesTraits65,66 for each set of 100 time-scaled input trees.

Line 549: You considered more alternative phylogenies besides of dropping Mesosaurus tenuidens. What's about a table which phylogenies were considered.

The alternative topological rearrangements of extinct marine reptiles are based on phylogenies provided by ref. 78, 79, and 80 as mentioned in the manuscript (see lines 568-569). We have now added these references also to Supplementary Table S2.

Supplement

Table S1: please provide the information extinct/extant

Revised.

Reviewer #3 (Remarks to the Author):

This draft is nearly ready for publication. I had reviewed at least two previous versions, for Nature. I found two sentences that need to be toned down to avoid sounding like you over-interpret your findings. Readers risk becoming suspicious if they get the impression that you want to sell moderate evidence as strong evidence. One sentence needs corrections and probably should be split, and another one contains a wrong taxonomic statement (see below). All this should take only minutes to fix.

We are grateful to Michel Laurin for his constant help.

Lines 20-21: "We show that non-EER oviparity evolved multiple times, and viviparity was the primitive reproductive..." To "show" is a strong statement in science. I would start this sentence by "Our analyses suggest that", or "Our analyses strongly favor..."

Revised

Line 41: “Strong evidence for the EER model is the near absence of fossils of...”. No, negative evidence plays a large role here (no evidence of the reproductive mode of any of the dozens of species of Paleozoic terrestrial amniotes), so at least remove the “strong”. I would put “circumstantial” there, or “weak”.

Revised

Line 47: “As EER occurs widely among extant lizards and snakes (lepidosauromorphs)”. Actually, “lizards” and snakes form Squamata, not Lepidosauromorpha, which is a much larger clade (with Sphenodon and many extinct taxa). Also, in such a paper, I would avoid paraphyletic taxa; why not just state “squamates”? I don’t think that most potential readers of this paper will be puzzled by this word.

Revised

Lines 134-138: “It could be argued that the described choristodere specimen is an incomplete viviparous egg^{27,28}, but in extant squamates, the egg of viviparous species lacking the calcified layer, but is enveloped with a very thin organic layer (commonly less than 10 μm), whereas oviparous squamates lay eggs with a calcified outer layer and a relatively thick organic layer (usually over 30 μm)^{29,30}, as here. » This sentence is too long, has two « but », and has a grammatical mistake “lack” is needed, rather than “lacking”.

Revised

Best wishes,

Michel Laurin

Reviewer #4 (Remarks to the Author):

25This paper reports two main results: the description of a fossil embryo/fetus with eggshell, and a phylogenetic analysis of parity mode. The fossil is identified as a choristodere embryo of *Ikechosauros*. I can not attest to the quality of this part of the paper since I am not a paleontologist, but given the expertise of the authors, I am willing accept their conclusions on the identity of the embryo. The phylogenetic analysis strongly suggests that extended embryo retention (EER) and may be even viviparity was the ancestral amniote reproductive mode. This conclusion is in contrast to the standard model which claims that the characteristic traits of the amniote egg are adaptations to terrestrial oviparity. The EER model is a very interesting novel development in our understanding of vertebrate reproductive biology and has been a minority position for most of the 20th century. If confirmed a model where the amniotic extraembryonic membranes evolved during egg retention would completely change the story about the functional significance of these structures. In a way, such a model would be rather attractive, because dropping an an-amniotic egg on land and hoping for the amnion to evolve to protect the embryo, does not seem to be a winning strategy. Teaching this model over the years I always felt unhappy, given that anyone who ever had alligator eggs in his/her lab knows that alligator eggs are NOT desiccation resistant, to the contrary!

Many thanks for the positive summary.

But if amniote egg structures evolved for fetal-maternal communication, this would make a lot of sense. It would also explain why even avian extra-embryonic tissues are producing steroid hormones and other signals, which could be residues of the time when they had to communicate with the mother during extended embryo retention. This is a potentially important paper, but see major comment 1 below.

Many thanks for the further positive remarks.

Major comments:

While the phylogenetic analysis is very well done and conservative in its assumptions, it is curious that on the synapsid side of the amniote phylogeny there are no non-mammalian fossils represented, as far as I can see. This could be due to a lack of developmental stages in the pre-mammalian synapsid fossil record, and I do not know whether that is the case. At the very least I would encourage the authors to

26state what is and is not known from pre-mammalian synapsid fossils. This is an important issue, because the current taxon sampling is heavily biased in favor of Reptilia/Sauropsida, which could give the impression of strong support of what is a reptilian rather than an amniote ancestral characteristic.

The absence of extinct synapsids in the analysis is that we could find no reliable examples with evidence of eggs or reproductive mode. We now state this in the introductory paragraph of 'Macroevolutionary study'.

2) The description of the novel embryo is only loosely related to the main result of the paper, which is based on the phylogenetic analysis of previously published data. The new fossil is way too young to be of significance for the phylogenetic analysis. Of course it is nice to have statistical inferences and real data combined in a paper, but I just want to point out that the connection between the two parts of the paper is not very strong.

The new specimen confirms oviparity existed in an assumed viviparous extinct clades, based on comparison with earlier descriptions of reproduction in the same genus. This demonstrates that evolutionarily labile reproductive strategy across oviparity to viviparity (EER), only known from squamates in extant amniotes, existed in basal archosauromorphs, potentially as well as in other various aquatic vertebrates of the past, such as mesosaurs, ichthyosaurs, and some sauropterygians. This suggests EER may be common in basal amniotes. We clarified this in the main text.

Minor comments:

This paper is very well written and cites older and even obscure but relevant literature, and I applaud the authors for that.

Thank you. We like to dig deep – and of course give credit properly to earlier work, even if more than 20 years old!

Günter Wagner, Yale University

Decision Letter, first revision:

22nd March 2023

Dear Dr. Jiang,

Thank you for submitting your revised manuscript "Extended embryo retention and viviparity in the first amniotes" (NATECOLEVOL-221117892A). It has now been seen again by the original reviewers and their comments are below. The reviewers find that the paper has improved in revision, and therefore we'll be happy in principle to publish it in Nature Ecology & Evolution, pending minor revisions to satisfy the reviewers' final requests and to comply with our editorial and formatting guidelines.

[REDACTED]

Reviewer #1 (Remarks to the Author):

The authors have dealt with the issues I raised in the first review

Reviewer #2 (Remarks to the Author):

I'm satisfied with the changes the authors made, as well as their responses (including the new results) to my previous comments. I recommend the paper to be accepted.

When going through the manuscript I spotted a few things that the authors should change before it is published.

Line 90: computed toMography

28Line 91: ... except for the ...

Line 181: single structured Markov model (SMM)

Line 185: different models and optimisation criteria. That's very unprecise. You use different methods on anstral state reconstruction, different evolutionary models and optimization methods.

Line 204: The SMM results ... would be more precise

Line 214: Using maximum parsimony (MP)-based ancestral state reconstruction instead of SMM did not change the main conclusions: ...

Line 218: That's the first use of an undefined acronym (ACCTTRAN)

Line 241: more simple ML, simple in terms of what?

Line 502: ... model considered different models on rate transition ...

Line 517: Please find a better word for uncertain. I thought that it is clear that the species has oviparous and viviparous individuals.

Line 531-532: ancestral state reconstruction for what/ which traits?

Line 535: amalgamated character (reproductive mode and mineralization)

Our ref: NATECOLEVOL-221117892A

27th March 2023

Dear Dr. Jiang,

Thank you for your patience as we've prepared the guidelines for final submission of your Nature Ecology & Evolution manuscript, "Extended embryo retention and viviparity in the first amniotes" (NATECOLEVOL-221117892A). Please carefully follow the step-by-step instructions provided in the attached file, and add a response in each row of the table to indicate the changes that you have made. Please also check and comment on any additional marked-up edits we have proposed within the text. Ensuring that each point is addressed will help to ensure that your revised manuscript can be swiftly handed over to our production team.

****We would like to start working on your revised paper, with all of the requested files and forms, as soon as possible (preferably within two weeks). Please get in contact with us immediately if you anticipate it taking more than two weeks to submit these revised files.****

29If you have not done so already, please alert us to any related manuscripts from your group that are under consideration or in press at other journals, or are being written up for submission to other journals (see: <https://www.nature.com/nature-research/editorial-policies/plagiarism#policy-on-duplicate-publication> for details).

In recognition of the time and expertise our reviewers provide to Nature Ecology & Evolution's editorial process, we would like to formally acknowledge their contribution to the external peer review of your manuscript entitled "Extended embryo retention and viviparity in the first amniotes". For those reviewers who give their assent, we will be publishing their names alongside the published article.

Nature Ecology & Evolution offers a Transparent Peer Review option for new original research manuscripts submitted after December 1st, 2019. As part of this initiative, we encourage our authors to support increased transparency into the peer review process by agreeing to have the reviewer comments, author rebuttal letters, and editorial decision letters published as a Supplementary item. When you submit your final files please clearly state in your cover letter whether or not you would like to participate in this initiative. Please note that failure to state your preference will result in delays in accepting your manuscript for publication.

Cover suggestions

As you prepare your final files we encourage you to consider whether you have any images or illustrations that may be appropriate for use on the cover of Nature Ecology & Evolution.

Nature Ecology & Evolution has now transitioned to a unified Rights Collection system which will allow our Author Services team to quickly and easily collect the rights and permissions required to publish your work. Approximately 10 days after your paper is formally accepted, you will receive an email in providing you with a link to complete the grant of rights. If your paper is eligible for Open Access, our Author Services team will also be in touch regarding any additional information that may be required to arrange payment for your article.

Please note that *Nature Ecology & Evolution* is a Transformative Journal (TJ). Authors may publish their research with us through the traditional subscription access route or make their paper immediately open access through payment of an article-processing charge (APC). Authors will not be required to make a final decision about access to their article until it has been accepted. [Find out more about Transformative Journals](https://www.springernature.com/gp/open-research/transformative-journals)

Authors may need to take specific actions to achieve [compliance with funder and institutional open access mandates](https://www.springernature.com/gp/open-research/funding/policy-compliance-faqs). If your research is supported by a funder that requires immediate open access (e.g. according to [Plan S principles](https://www.springernature.com/gp/open-research/plan-s-compliance)) then you should select the gold OA route, and we will direct you to the compliant route where possible. For authors selecting the subscription publication route, the journal's standard licensing terms will need to be accepted, including [self-archiving-and-license-to-publish](https://www.nature.com/nature-portfolio/editorial-policies/self-archiving-and-license-to-publish). Those licensing terms will supersede any other terms that the author or any third party may assert apply to any version of the manuscript.

Please use the following link for uploading these materials:
[REDACTED]

[REDACTED]

Reviewer #1:

Remarks to the Author:

The authors have dealt with the issues I raised in the first review

Reviewer #2:

Remarks to the Author:

I'm satisfied with the changes the authors made, as well as their responses (including the new results) to my previous comments. I recommend the paper to be accepted.

When going through the manuscript I spotted a few things that the authors should change before it is published.

Line 90: computed toMography

Line 91: ... except for the ...

Line 181: single structured Markov model (SMM)

Line 185: different models and optimisation criteria. That's very unprecise. You use different methods on anstral state reconstruction, different evolutionary models and optimization methods.

Line 204: The SMM results ... would be more precise

Line 214: Using maximum parsimony (MP)-based ancestral state reconstruction instead of SMM did not change the main conclusions: ...

Line 218: That's the first use of an undefined acronym (ACCTTRAN)

Line 241: more simple ML, simple in terms of what?

Line 502: ... model considered different models on rate transition ...

Line 517: Please find a better word for uncertain. I thought that it is clear that the species has oviparous and viviparous individuals.

Line 531-532: ancestral state reconstruction for what/ which traits?

Line 535: amalgamated character (reproductive mode and mineralization)

Author Rebuttal, first revision:

Response to reviewer' comments

Reviewer #1 (Remarks to the Author):

The authors have dealt with the issues I raised in the first review

Thanks!

Reviewer #2 (Remarks to the Author):

I'm satisfied with the changes the authors made, as well as their responses (including the new results) to my previous comments. I recommend the paper to be accepted.

Thanks for the recognition of our efforts, as well these correction suggestions!

32When going through the manuscript I spotted a few things that the authors should change before it is published.

Line 90: computed tomography

Corrected.

Line 91: ... except for the ...

Corrected.

Line 181: single structured Markov model (SMM)

Corrected.

Line 185: different models and optimisation criteria. That's very unprecise. You use different methods on anstral state reconstruction, different evolutionary models and optimization methods.

Changed to:

“iv) different ancestral state reconstruction methods, evolutionary models and optimization criteria;”

Line 204: The SMM results ... would be more precise

Changed to:

“The ancestral state reconstruction results are unequivocal for both the amalgamated character (reproduction mode + eggshell mineralisation) and EER presence.”

Note, that SMMs were only used for the amalgamated character, not for modelling the EER presence/absence since there is no character dependency to be modelled for this two-state character.

Line 214: Using maximum parsimony (MP)-based ancestral state reconstruction instead of SMM did not change the main conclusions: ...

Kept the original wording:

“Using a different ancestral state reconstruction method also does not change the main conclusions: [...]”

The same SMMs were used both for maximum likelihood and Bayesian analyses. For parsimony we used an approach relying on Sankoff matrices that is akin to SMM. It is not exactly the same due to the nature of parsimony analyses (see the methods section of our main text for a more detailed explanation) but technically it would not be correct to say that we used MP instead of SMM.

Line 218: That's the first use of an undefined acronym (ACCTTRAN)

Changed to:

“all MP-based reconstructions using accelerated transformation (ACCTTRAN) favour [...]”

Line 241: more simple ML, simple in terms of what?

Changed to:

“If we also consider more simple ML models with a worse fit (AIC difference > 2), where character state transition rates are more constrained compared to the best-fitting models, the evidence for parchment-shelled eggs for these clades increases.”

Line 502: ... model considered different models on rate transition ...

Corrected.

Line 517: Please find a better word for uncertain. I thought that it is clear that the species has oviparous and viviparous individuals.

Changed to:

“Current fossil evidence does not allow to determine confidently whether *Mesosaurus tenuidens* was viviparous or oviparous with membrane-shelled eggs^{14,17}.”

Line 531-532: ancestral state reconstruction for what/ which traits?

Changed to:

“In addition to ML, we also ran a parsimony (MP) based ancestral state reconstruction for the same characters using [...]”

Line 535: amalgamated character (reproductive mode and mineralization)

Corrected.

Final Decision Letter:

17th April 2023

Dear Professor Jiang,

We are pleased to inform you that your Article entitled "Extended embryo retention and viviparity in the first amniotes", has now been accepted for publication in Nature Ecology & Evolution.

Over the next few weeks, your paper will be copyedited to ensure that it conforms to Nature Ecology and Evolution style. Once your paper is typeset, you will receive an email with a link to choose the appropriate publishing options for your paper and our Author Services team will be in touch regarding any additional information that may be required

You will not receive your proofs until the publishing agreement has been received through our system

Due to the importance of these deadlines, we ask you please us know now whether you will be difficult to contact over the next month. If this is the case, we ask you provide us with the contact information (email, phone and fax) of someone who will be able to check the proofs on your behalf, and who will

35be available to address any last-minute problems. Once your paper has been scheduled for online publication, the Nature press office will be in touch to confirm the details.

Acceptance of your manuscript is conditional on all authors' agreement with our publication policies (see www.nature.com/authors/policies/index.html). In particular your manuscript must not be published elsewhere and there must be no announcement of the work to any media outlet until the publication date (the day on which it is uploaded onto our web site).

Please note that *Nature Ecology & Evolution* is a Transformative Journal (TJ). Authors may publish their research with us through the traditional subscription access route or make their paper immediately open access through payment of an article-processing charge (APC). Authors will not be required to make a final decision about access to their article until it has been accepted. [Find out more about Transformative Journals](https://www.springernature.com/gp/open-research/transformative-journals)

Authors may need to take specific actions to achieve [compliance with funder and institutional open access mandates](https://www.springernature.com/gp/open-research/funding/policy-compliance-faqs). If your research is supported by a funder that requires immediate open access (e.g. according to [Plan S principles](https://www.springernature.com/gp/open-research/plan-s-compliance)) then you should select the gold OA route, and we will direct you to the compliant route where possible. For authors selecting the subscription publication route, the journal's standard licensing terms will need to be accepted, including [self-archiving and license to publish](https://www.nature.com/nature-portfolio/editorial-policies/self-archiving-and-license-to-publish). Those licensing terms will supersede any other terms that the author or any third party may assert apply to any version of the manuscript.

We welcome the submission of potential cover material (including a short caption of around 40 words) related to your manuscript; suggestions should be sent to Nature Ecology & Evolution as electronic files (the image should be 300 dpi at 210 x 297 mm in either TIFF or JPEG format). Please note that such pictures should be selected more for their aesthetic appeal than for their scientific content, and

36that colour images work better than black and white or grayscale images. Please do not try to design a cover with the Nature Ecology & Evolution logo etc., and please do not submit composites of images related to your work. I am sure you will understand that we cannot make any promise as to whether any of your suggestions might be selected for the cover of the journal.

You can generate the link yourself when you receive your article DOI by entering it here: <http://authors.springernature.com/share>.

[REDACTED]

P.S. Click on the following link if you would like to recommend Nature Ecology & Evolution to your librarian <http://www.nature.com/subscriptions/recommend.html#forms>

** Visit the Springer Nature Editorial and Publishing website at http://editorial-jobs.springernature.com?utm_source=ejp_NEcoE_email&utm_medium=ejp_NEcoE_email&utm_campaign=ejp_NEcoE for more information about our career opportunities. If you have any questions please click [here](mailto:editorial.publishing.jobs@springernature.com). **